# Elucidated Rolling Diffusion Models for Probabilistic Forecasting of Complex Dynamics

**Salva Rühling Cachay**
UC San Diego

**Miika Aittala**
NVIDIA

**Karsten Kreis**
NVIDIA

**Noah Brenowitz**
NVIDIA

**Arash Vahdat**
NVIDIA

**Morteza Mardani**[*]
NVIDIA

**Rose Yu**[*]
UC San Diego

## Abstract

Diffusion models are a powerful tool for probabilistic forecasting, yet most applications in high-dimensional complex systems predict future states individually. This approach struggles to model complex temporal dependencies and fails to explicitly account for the progressive growth of uncertainty inherent to the systems. While rolling diffusion frameworks, which apply increasing noise to forecasts at longer lead times, have been proposed to address this, their integration with state-of-the-art, high-fidelity diffusion techniques remains a significant challenge. We tackle this problem by introducing Elucidated Rolling Diffusion Models (ERDM), the first framework to successfully unify a rolling forecast structure with the principled, performant design of Elucidated Diffusion Models (EDM). To do this, we adapt the core EDM components–its noise schedule, network preconditioning, and Heun sampler–to the rolling forecast setting. The success of this integration is driven by three key contributions: $(i)$ a novel loss weighting scheme that focuses model capacity on the mid-range forecast horizons where determinism gives way to stochasticity; $(ii)$ an efficient initialization strategy using a pre-trained EDM for the initial window; and $(iii)$ a bespoke hybrid sequence architecture for robust spatiotemporal feature extraction under progressive denoising. On 2D Navier–Stokes simulations and ERA5 global weather forecasting at $1.5°$ resolution, ERDM consistently outperforms key diffusion-based baselines, including conditional autoregressive EDM. ERDM offers a flexible and powerful general framework for tackling diffusion-based dynamics forecasting problems where modeling uncertainty propagation is paramount.[1]

## 1 Introduction

Probabilistic forecasting of complex dynamical systems is essential for realistically assessing future states ("snapshots") and their inherent uncertainties [2, 11]. For example, medium-range weather forecasting ($\leqslant 15$ days) grapples with the inherent chaotic nature of the atmosphere and high sensitivities to the initial conditions. Ensembles of numerical weather prediction (NWP) models are therefore standard practice for estimating forecast uncertainty and the likelihood of high-impact events [42, 3]. However, each ensemble run of a flagship system such as IFS ENS [15] is computationally expensive, motivating data-driven alternatives [7].

---

[1]Code is available at: https://github.com/NVlabs/ERDM

[*]Equal advising. Corresponding authors listed alphabetically.

39th Conference on Neural Information Processing Systems (NeurIPS 2025).

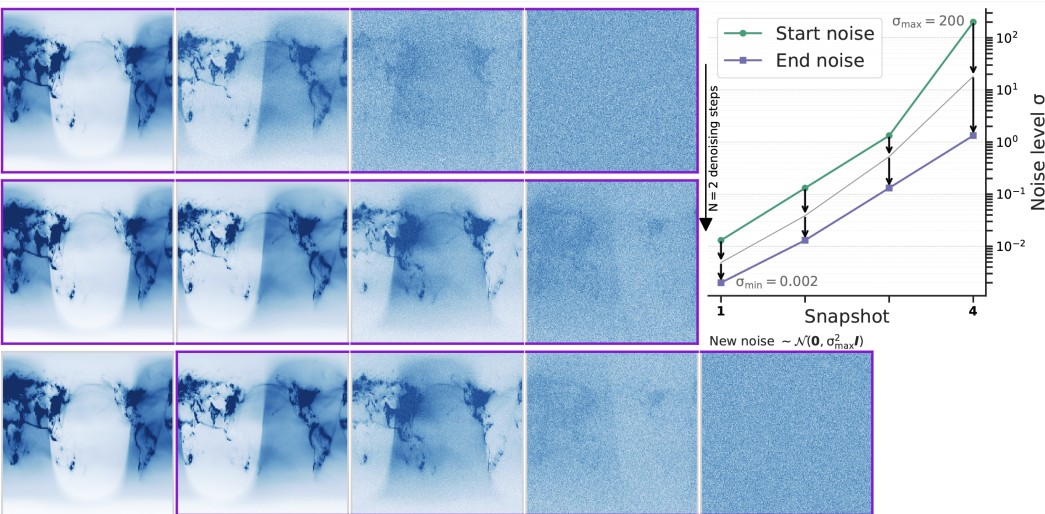

Figure 1: ERDM sampling with a window, highlighted in bold purple, of size $W = 4$. **Top row:** ERDM starts at diffusion time $t = 0$ with snapshots, $x_1, \ldots, x_W$, corrupted by progressively larger noise levels, $\bar{\sigma}_1(t) < \cdots < \bar{\sigma}_W(t) = \sigma_{\max}$. **Middle row:** After $N = 2$ joint denoising steps, the sequence reaches lower noise levels at $t = 1$ such that $\bar{\sigma}_1(1) = \sigma_{\min}$ and $\bar{\sigma}_w(1) = \bar{\sigma}_{w-1}(0)$ for $w > 1$, as illustrated in the right-hand panel. The now fully denoised first snapshot, $x_1$, is returned. **Bottom row:** The rest of the sequence is shifted one slot to the right, and a fresh pure-noise snapshot is appended to the new window. The cycle then repeats.

Diffusion models have shown notable success in generative modeling, particularly for high-dimensional data such as images and videos [52, 13]. The EDM framework [26], for example, enables stable high-fidelity sample generation due to its principled design that generalizes and improves on the foundational denoising diffusion probabilistic model (DDPM) [21] and related diffusion paradigms [53, 54]. This principled approach makes them especially promising for modeling time-evolving dynamical systems. In this domain, the ability to generate sharp, diverse ensembles is crucial for robust uncertainty quantification, and producing high-fidelity, physically consistent samples is a key advantage over deterministic models that often yield blurry, averaged-out predictions.

Autoregressive conditional diffusion models have been successfully applied to fluid dynamics [29] and weather [45]. These approaches often use limited temporal context and predict future snapshots one by one. To explicitly model the progressively increasing uncertainty common to sequence generation tasks, Rolling Sequence Diffusion Models (RSDM) [49, 57] employ snapshot-dependent noise schedules that escalate noise for distant future snapshots. However, existing RSDM's are derived from DDPM, rather than leveraging improved practical design choices, such as network preconditioning and second-order sampling, that were key to EDM's success.

Our work addresses these limitations by building upon the successful EDM framework [26]. We adapt and extend EDM's core design elements—its noise schedule formulation, denoiser network parameterization, network preconditioning, loss weighting, and sampling algorithms—to the specific demands of RSDM-like sequence modeling. In developing ERDM, we identified and addressed key design choices for progressive noise schedules and temporal loss weighting within the EDM paradigm, aspects that received limited attention in prior work. Our work thus provides novel solutions for these crucial components. All in all, our key contributions are summarized as follows:

- We develop **Elucidated Rolling Diffusion Models (ERDM)**, integrating snapshot-dependent progressive noise levels into EDM, by adapting EDM's *noise schedule, loss weighting, preconditioning*, and *sampling*, complemented by a simple, but effective first-window initialization technique and a hybrid 3D denoiser architecture.

- Extensive experiments on ERA5 weather data and Navier-Stokes flows show that ERDM consistently improves on the autoregressive, conditional EDM baseline in both predictive skill and calibration. On ERA5, ERDM rivals state-of-the-art weather forecasters (IFS ENS and NeuralGCM ENS) for mid- to long-range lead times, while being computationally more efficient–requiring only 4 H200 GPUs and 5 days for training–and exhibiting high physical fidelity in its power spectra.

## 2 Related Works

**Diffusion models for spatiotemporal data.** Diffusion models, initially transformative in image and audio generation, are increasingly being adapted for spatiotemporal modeling [58, 20, 17, 50]. Latent-diffusion variants [56, 47] provide the computational advantages integral to state-of-the-art video generators [22, 5]. Recent work refines the diffusion process itself to strengthen temporal structure and boost efficiency [50, 32]. Rolling sequence diffusion models (RSDM) advance this idea by assigning progressively larger noise levels to later timesteps, mirroring the growth of predictive uncertainty [57, 49, 27]. We embrace the same intuition but depart from these DDPM-based RSDM's in two crucial ways. (i) Our approach is derived from the EDM framework, which generalises and improves upon DDPM. (ii) We re-examine often-overlooked design choices—noise schedule, loss weighting, and network architecture—and show that their coordination is critical for forecasting complex dynamics. Progressive schedules like ours form a specific instance of the completely randomized schedules proposed in Chen et al. [9], corresponding to its 'pyramid sampling' scheme. However, our ablations indicate that such training-time randomization degrades performance–possibly because it misaligns with the temporally increasing uncertainty inherent to the chaotic systems that we study.

**Data-driven medium-range weather forecasting.** Initial machine learning (ML)-based attempts in this area were limited to deterministic forecasting [43, 4, 30, 38, 39, 6], often differentiated by their underlying neural network architectures [25]. More recently diffusion and flow matching models have shown promise for probabilistic weather forecasting [45, 12, 1], and other tasks such as downscaling [36, 34, 10, 55], data assimilation [23, 48, 35], and emulation [31, 51, 44]. Latent-variable models offer an alternative probabilistic route [40], and physics-ML hybrids such as NeuralGCM show promise but face scalability constraints [28]. Of particular relevance to our work is GenCast [45], which successfully applied the EDM framework, using the graph architecture from [30], for next-step probabilistic weather forecasting. Our primary baseline in this paper, EDM, corresponds to the same approach but using our experimental setup (e.g., same U-Net architecture and dataset resolution; see Appendix D.4 for more details). This design ensures a controlled and direct comparison between ERDM and an EDM-based next-step forecasting approach.

## 3 Background

In the following, we refer to clean data as $y$, noisy data as $x$, and we use $\bar{x}$ to make clear when the data is corrupted according to progressively increasing noise levels. We abbreviate sequences by $y_{1:W} := y_1, \ldots, y_W$. Probabilistic forecasting is concerned with predicting a future sequence $\{y_\tau\}_{\tau=1}^{T_{\text{forecast}}}$ given an initial condition $y_0$, where the horizon $T_{\text{forecast}}$ can be arbitrarily large. This requires learning the joint conditional distribution, which can be approximated with, e.g., diffusion models, by $p_\theta(y_{1:T_{\text{forecast}}}|y_0)$. Directly modeling the high-dimensional joint distribution is often intractable. A common alternative is modeling a smaller window of size $W$ by learning a short-term transition, $p_\theta(y_{w+1:w+W}|y_w)$ [45, 20]. Long sequences are then generated autoregressively.

Conditional diffusion models can be applied to this task by learning a denoiser network $D_\theta(x_{1:W}; \sigma, y_0)$. The denoiser is trained to predict the target sequence, $y_{1:W}$, given the initial condition, $y_0$, and data $x_{1:W}$ corrupted with Gaussian noise of standard deviation $\sigma \in [\sigma_{\min}, \sigma_{\max}]$. The maximum noise level needs to be chosen such that $\sigma_{\max} \gg \sigma_{\text{data}}$, where $\sigma_{\text{data}}$ is the standard deviation of the data $y \sim p_{\text{data}}$ and $y \in \mathbb{R}^D$. The trained network can then be used jointly with an ODE or SDE solver to iteratively generate the clean data starting from pure noise, $\epsilon \sim \mathcal{N}(0, \sigma_{\max}^2 \mathbf{I}_{W \times D})$, and $y_0$. See Appendix B for an additional background on diffusion. Often, $W = 1$ [45, 29], and the diffusion model becomes a next-step forecasting model. However, the short window size models *limited temporal interactions* and generating each step requires a full reverse diffusion process, incurring substantial *computational cost*. Standard sequence diffusion models with $W > 1$ typically apply noise uniformly across the prediction window, failing to capture the *increasing uncertainty* inherent to future predictions.

Rolling Sequence Diffusion Models (RSDMs) [49, 9] address prior limitations by explicitly incorporating progressive forecasting uncertainty. The core innovation is a *progressive noise schedule* with $W > 1$. Instead of uniform noise, RSDMs use a monotonically increasing schedule of noise standard deviations $0 \approx \bar{\sigma}_1(t) < \bar{\sigma}_2(t) < \cdots < \bar{\sigma}_W(t) \approx \sigma_{\max}$, where $t$ denotes the global diffusion

time. A noisy future state is thus modeled as $\bar{\boldsymbol{x}}_w \sim \mathcal{N}(\boldsymbol{y}_w, \bar{\sigma}_w^2(t)\mathbf{I}_D)$. This structure reflects higher uncertainty for predictions further into the future. The denoiser $D_\theta(\bar{\boldsymbol{x}}_{1:W}; \bar{\boldsymbol{\sigma}}_{1:W}(t))$ is trained analogously to a standard diffusion model to predict the clean sequence, $\boldsymbol{y}_{1:W}$. In our implementation, we omit direct conditioning of the denoiser on the last known clean state, $\boldsymbol{y}_0$. This is feasible because we initialize the first window using an external forecaster (itself conditioned on $\boldsymbol{y}_0$). The minimally noisy start of the window provides sufficient conditioning to predict subsequent snapshots. When requiring the RSDM to self-initialize its first window, this is not possible [49].

## 4  Forecasting with Elucidated Rolling Diffusion Models

Our work, ERDM, integrates the rolling diffusion concept with the Elucidated Diffusion Model (EDM) framework [26]. While RSDMs introduce the idea of progressive noise, they usually do not follow EDM's principled design choices for preconditioning, loss weighting, and sampler design, which we hypothesize would similarly improve RSDM performance. ERDM aims to systematically port and adapt these EDM principles to the rolling, progressive noise setting, thereby providing a robust and theoretically grounded approach to sequential data generation.

**Rolling EDM noise schedule.**  In contrast to existing RSDMs, which use cosine or linear noise schedules, we propose adapting EDM's sampling-time noise schedule to capture progressively increasing noise levels, by defining

$$\bar{\boldsymbol{\sigma}}(t) = (\bar{\sigma}_1(t), \ldots, \bar{\sigma}_W(t)); \qquad \bar{\sigma}_w(t) = \left(\sigma_{\max}^{\frac{1}{\rho}} + t_{w,t}(\sigma_{\min}^{\frac{1}{\rho}} - \sigma_{\max}^{\frac{1}{\rho}})\right)^\rho, \qquad (1)$$

where $t_{w,t} = 1 - \frac{w-t}{W}$ is the local diffusion time of the snapshot. The $1-$ is important so that near-future snapshots receive minimal amounts of noise, while far-future snapshots receive high levels of noise. This ensures the correct inductive bias that distant snapshots are more uncertain than near-future ones. The continuous noise schedule is effectively divided into $W$ segments such that the noise levels at snapshot $w$ satisfy $\bar{\sigma}_w(t) \in [\bar{\sigma}_{w-1}(0), \bar{\sigma}_{w+1}(1)]$, where $t \in [0,1]$, $\bar{\sigma}_0(0) := \sigma_{\min}$, and $\bar{\sigma}_{W+1}(1) := \sigma_{\max}$. Our formulation thus satisfies $\sigma_{\min} = \bar{\sigma}_1(1) < \bar{\sigma}_1(0) = \bar{\sigma}_2(1) < \cdots < \bar{\sigma}_W(0) = \sigma_{\max}$. Two example noise schedules are illustrated in Fig. 2. ERDM is trained to see any noise schedule level within the noise segments (color gradients in Fig. 2) by randomizing $t \in U([0,1))$. During sampling, each $\bar{\sigma}_w(t)$ is evolved from $\bar{\sigma}_w(0)$ to $\bar{\sigma}_w(1)$. We found it important to tune the curvature parameter, $\rho$. Our proposed de-

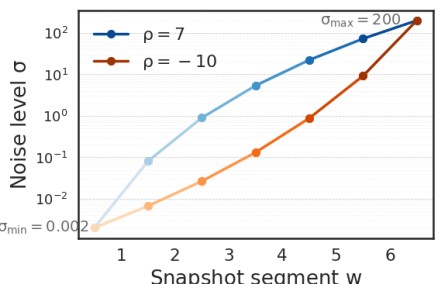

Figure 2: Noise schedule comparison for a sequence length $W = 6$, yielding 6 visualized segments $[\bar{\sigma}_w(1), \bar{\sigma}_w(0)]$. The schedules differ by their curvature parameter $\rho$: one using $\rho = 7$ (default EDM) and the other $\rho = -10$ (ERDM). Color gradients illustrate segment progression.

fault value, $\rho = -10$, effectively causes snapshots to be under less noise compared to the default EDM choice, $\rho = 7$, which provides more information for accurate denoising. Importantly, these noise schedule parameters $\sigma_{\min}, \sigma_{\max}$, and $\rho$ need to be chosen at training time.

**Probability flow ODE.**  Having defined our windowed noise schedule $\bar{\boldsymbol{\sigma}}(t) = (\bar{\sigma}_1(t), \ldots, \bar{\sigma}_W(t))$, we can formulate the associated probability flow ODE [54], which describes the addition and removal of noise when moving the diffusion 'time', $t$, forward $(1 \rightarrow 0)$ and backward $(0 \rightarrow 1)$, respectively. For a noisy sequence $\bar{\boldsymbol{x}} := \bar{\boldsymbol{x}}_{1:W}$, this ODE, as derived in Appendix C.1, is:

$$d\bar{\boldsymbol{x}} = -\text{diag}(\bar{\sigma}_1(t)\dot{\sigma}_1(t)\mathbf{I}_D, \ldots, \bar{\sigma}_W(t)\dot{\sigma}_W(t)\mathbf{I}_D)\nabla_{\bar{\boldsymbol{x}}} \log p(\bar{\boldsymbol{x}}; \bar{\boldsymbol{\sigma}}(t))dt, \qquad (2)$$

where $\dot{\sigma}_w(t)$ denotes the diffusion time derivative of $\bar{\sigma}_w(t)$, and $\nabla_{\bar{\boldsymbol{x}}} \log p(\bar{\boldsymbol{x}}; \bar{\boldsymbol{\sigma}}(t))$ is the score function [24] that we will approximate with a denoiser network. This ODE governs $W$ coupled diffusion processes evolving simultaneously within the window. During inference (i.e., the backward process), an ODE solver is used, typically for N discrete steps. Upon completion of one such integration to $t = 1$, the first snapshot, $\bar{\boldsymbol{x}}_1(1)$, is fully denoised (its noise level $\bar{\sigma}_1(1) = \sigma_{\min}$) and is output as the forecast of $\boldsymbol{y}_1$. Concurrently, the subsequent snapshots $\bar{\boldsymbol{x}}_{2:W}(1)$ are partially denoised. Key to the rolling mechanism is the design of the per-snapshot noise schedules $\bar{\sigma}_w(t)$, which ensure the noise level of $\bar{\boldsymbol{x}}_w(1)$ matches the initial noise level of the $(w-1)$-th slot (i.e., $\bar{\sigma}_w(1) = \bar{\sigma}_{w-1}(0)$).

This specific noise structure at $t = 1$ enables the window to be advanced efficiently. For the next forecasting step, the sequence $\bar{\boldsymbol{x}}_{2:W}(1)$ forms the initial state for the first $W - 1$ slots at the next iteration's $t = 0$. A new, fully-noised snapshot, $\boldsymbol{\epsilon}_{W+1} \sim \mathcal{N}(\mathbf{0}, \sigma_{\max}^2 \mathbf{I}_D)$, is then appended as the $W$-th element of this new window. The ODE solving process from $t = 0$ to $1$ is then repeated for this updated window. This iterative sampling procedure is visualized in Fig. 1.

## 4.1 Training Objective

We now describe how to train a denoiser network, $D_\theta(\bar{\boldsymbol{x}}; \bar{\boldsymbol{\sigma}}(t))$, that can be used to solve the probability flow ODE above. Following EDM, it comprises a raw neural network $F_\theta$ and preconditioning. To account for snapshot-dependent noise levels, we vectorize the preconditioning functions:

$$D_\theta(\bar{\boldsymbol{x}}; \bar{\boldsymbol{\sigma}}(t)) = c_{\text{skip}}(\bar{\boldsymbol{\sigma}}(t))\bar{\boldsymbol{x}} + c_{\text{out}}(\bar{\boldsymbol{\sigma}}(t))F_\theta(c_{\text{in}}(\bar{\boldsymbol{\sigma}}(t))\bar{\boldsymbol{x}}, c_{\text{noise}}(\bar{\boldsymbol{\sigma}}(t))). \tag{3}$$

The preconditioning functions $c_{\text{skip}}, c_{\text{out}}, c_{\text{in}}, c_{\text{noise}}$ are adapted from EDM and applied per-snapshot based on the corresponding $\bar{\sigma}_w(t)$ in $\bar{\boldsymbol{\sigma}}(t)$. The denoiser network is trained to predict the clean window $\boldsymbol{y}_{1:W}$ from its noisy version, $\bar{\boldsymbol{x}} \sim \mathcal{N}(\boldsymbol{y}_{1:W}, \bar{\boldsymbol{\sigma}}(t)^2\mathbf{I})$. During training, we randomize $t \sim U([0,1))$ such that the network learns to deal with the full extent of the segments of Fig. 2.

**Uncertainty-aware loss reweighting.** EDM employs a loss weighting $\lambda(\sigma) = (\sigma^2 + \sigma_{\text{data}}^2)/(\sigma\sigma_{\text{data}})^2$ to ensure their core network $F_\theta$ targets a signal with unit variance, promoting training stability. We apply $\lambda(\bar{\sigma}_w)$ per frame as a direct extension. However, the rolling mechanism implies that certain noise levels within the $\bar{\boldsymbol{\sigma}}(t)$ window are more critical for learning the "de-mixing" of temporal information. EDM itself indirectly emphasizes intermediate noise levels by sampling $\sigma$ values from a lognormal distribution $p_{\text{train}}(\sigma)$ during training, whose PDF corresponds to

$$f(\sigma; P_{\text{mean}}, P_{\text{std}}) = \frac{1}{\sigma P_{\text{std}}\sqrt{2\pi}} \exp\left(-\frac{(\ln(\sigma) - P_{\text{mean}})^2}{2P_{\text{std}}^2}\right). \tag{4}$$

Since ERDM conditions on a fixed $\bar{\boldsymbol{\sigma}}(t)$ per training instance, always covering a wide range of noise levels (rather than sampling each $\bar{\sigma}_w$ from $p_{\text{train}}(\sigma)$), we propose a distinct loss weighting for ERDM that combines the EDM unit-variance objective with an emphasis on these critical intermediate noise levels within the progressive schedule. The effective loss weighting for snapshot $w$ in ERDM is $\lambda(\bar{\sigma}_w) \cdot f(\bar{\sigma}_w; P_{\text{mean}}, P_{\text{std}})$, which maintains EDM's target normalization via $\lambda(\bar{\sigma}_w)$ while using $f(\bar{\sigma}_w)$ to upweight snapshots at the most informative noise levels. All things considered, we can write out our proposed score matching objective as

$$\min_\theta \mathbb{E}_{\boldsymbol{y}_{1:W} \sim p_{\text{data}}} \mathbb{E}_{t \in U([0,1))} \mathbb{E}_{\{\boldsymbol{\epsilon}_w \sim \mathcal{N}(\mathbf{0}, \boldsymbol{\sigma}_w^2\mathbf{I})\}_{w=1}^W} \sum_{w=1}^W \lambda(\sigma_w)f(\sigma_w)\|D_\theta(\boldsymbol{y}_{1:W} + \boldsymbol{\epsilon}_{1:W}; \boldsymbol{\sigma})_w - \boldsymbol{y}_w\|_2^2,$$

where we shorten $\boldsymbol{\sigma} := \bar{\boldsymbol{\sigma}}(t)$. The full training algorithm is described in Algorithm 1.

---

**Algorithm 1** Elucidated Rolling Diffusion: Training

---

1: **Require:** Training data $\mathcal{D}_{\text{train}}$, network $F_\theta$, $\sigma_{\min}, \sigma_{\max}, \rho, P_{\text{mean}}, P_{\text{std}}$
2: **repeat**
3:      Sample $\boldsymbol{y} = (\boldsymbol{y}_1, \ldots, \boldsymbol{y}_W) \in \mathbb{R}^{W \times D}$ from $\mathcal{D}_{\text{train}}, \boldsymbol{\epsilon} \sim \mathcal{N}(\mathbf{0}, \mathbf{I}_{W \times D}), t \sim U([0,1))$
4:      $\boldsymbol{\sigma} = (\sigma_1, \ldots, \sigma_W) \leftarrow \bar{\boldsymbol{\sigma}}(t)$
5:      $\bar{\boldsymbol{x}} \leftarrow \boldsymbol{y} + \boldsymbol{\sigma} \cdot \boldsymbol{\epsilon}$                                            ▷ Add rolling noise to data
6:      $\hat{\boldsymbol{y}} \leftarrow c_{\text{skip}}(\boldsymbol{\sigma})\bar{\boldsymbol{x}} + c_{\text{out}}(\boldsymbol{\sigma})F_\theta(c_{\text{in}}(\boldsymbol{\sigma})\bar{\boldsymbol{x}}, c_{\text{noise}}(\boldsymbol{\sigma}))$
7:      Update $\theta$ using $L_\theta = \frac{1}{W}\sum_{w=1}^W \lambda(\sigma_w)f(\sigma_w; P_{\text{mean}}, P_{\text{std}})\|\boldsymbol{y}_w - \hat{\boldsymbol{y}}_w\|_2^2$
8: **until** Converged

---

**Remark (temporal noise correlations).** In both training and sampling algorithms, we draw i.i.d. Gaussian noise $\boldsymbol{\epsilon}_w \sim \mathcal{N}(\mathbf{0}, \mathbf{I}_D)$ for each snapshot $w$ independently. Recent studies show that temporally correlated noise improves video diffusion [8, 18, 1, 33]. Such priors are orthogonal to our sliding-window framework and can replace the i.i.d. draws in Algorithms 1–2. In our experiments, we use the progressive noise model with $\alpha = 1$ proposed by Ge et al. [18], which we found to slightly improve long-range forecasts (see Appendix E). More details are provided in Appendix C.4.

## 4.2 Sampling

The trained denoiser, $D_\theta$, can be used to estimate the score function, $\log p(\bar{x}; \bar{\sigma}(t))$, in Eq. (2) with $(D_\theta(\bar{x}, \bar{\sigma}(t)) - \bar{x})/\bar{\sigma}^2(t)$ for any $t \in [0, 1)$ [26]. Based on this identity and the described probability flow ODE intuition, a mathematical formulation of our proposed sampling algorithm is provided in Algorithm 2. For simplicity, we omit the second-order Heun step and sampling stochasticity, which are detailed in Appendix C.2. Note, however, that in our main experiments, we always use the deterministic Heun sampler.

**First-window initialization.** The backward ODE requires a noisy window $\bar{x}_{1:W}$ whose clean latent $y_{1:W}$ is unknown. Ruhe et al. [49] addresses this issue by co-training the rolling model to generate $\bar{x}_{1:W}$ from clean context snapshots and a sequence of pure noise. This approach introduces extra hyperparameters and diverts the denoiser's capacity between two distinct denoising tasks. As an alternative, we propose drawing $\hat{y}_{1:W}$ from an *external* forecaster $p_\theta(y_{1:W}|y_0)$. We then sample $\bar{x}_w \sim \mathcal{N}(\hat{y}_w, \bar{\sigma}_w^2(0)\mathbf{I}_D)$ for $w = 1, \ldots, W$, from where we can start our sliding-window sampling algorithm. This choice reuses mature short-range models and injects schedule-matched uncertainty; any existing forecaster (diffusion-based or not) can supply $\hat{y}_{1:W}$. In our experiments, we rely on the EDM baselines for initializing ERDM. We ablate this choice in Appendix E.1.

### 4.3 Denoiser architecture for temporal dynamics

Our proposed diffusion model operates on temporal sequences, necessitating an architecture capable of capturing temporal dependencies. Naive adaptations of 2D architectures, such as stacking the time dimension into channels, disrupt the inherent temporal structure and led to suboptimal results in our experiments (see Section 5.4). Alternatively, extending to 3D convolutions, while preserving temporality, often incurs significant computational overhead and potentially higher sample complexity. Instead, we adopt a prevalent strategy from latent video diffusion models [5]: augmenting a well-established 2D U-Net architecture with explicit temporal processing layers. Specifically, we integrate causal temporal attention layers into the 2D ADM U-Net [13] used in EDM, positioning before each down- and up-sampling block. These temporal layers also incorporate noise level information via a mechanism analogous to the adaptive layer normalization used in the 2D U-Net blocks. Figure 3 illustrates this interleaved modular design. This modular design could facilitate static pre-training of the 2D backbone, yet we observed that end-to-end training on the sequential data was more effective for our tasks (see Appendix E). While we found the proposed architecture to be key for generating good forecasts, it is slower and has higher memory needs than a 2D variant.

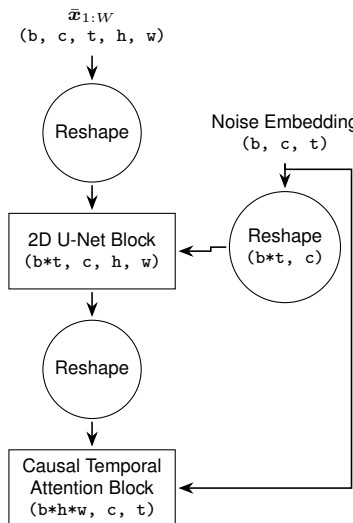

Figure 3: Sketch of one 2D U-Net block and one temporal attention block in our hybrid U-Net topology with noise embedding to both spatial and temporal paths. Dimensions `b`, `c`, `t`, `h`, `w` refer to batch, channel, window, height, and width. For simplicity, `c`, `h`, `w` do not change in the sketch.

## 5 Experiments

### 5.1 Evaluation and Metrics

Due to the importance of probabilistic forecasts and uncertainty quantification, we focus on two key ensemble-based probabilistic metrics: Continuous Ranked Probability Score (CRPS) [37], and the spread-skill ratio (SSR) based on $M$-member ensembles, where $M = 50$ for Navier-Stokes and $M = 10$ for ERA5. The CRPS is a proper scoring rule commonly used to evaluate probabilistic forecasts [19]. The spread-skill ratio is defined as the ratio of the square root of the ensemble variance to the corresponding ensemble-mean RMSE. It serves as a measure of the reliability of the ensemble, where values smaller than 1 indicate underdispersion, and larger values overdispersion [16, 46].

**Algorithm 2** Elucidated Rolling Diffusion Deterministic Sampler (Euler-only)

1: **Require:** $\hat{\boldsymbol{y}}_{1:W}, N, T_{\text{forecast}}$
2: $\Delta t \leftarrow 1/N$       ▷ Infer step size from desired number of steps per snapshot
3: $t_{\text{cur}} \leftarrow 0; \mathcal{S} \leftarrow \varnothing$       ▷ Initialize global diffusion time and empty generated sequence
4: **sample** $\bar{\boldsymbol{x}}_{\text{cur}} \sim \mathcal{N}(\hat{\boldsymbol{y}}_{1:W}, \bar{\boldsymbol{\sigma}}(t_{\text{cur}})^2 \mathbf{I}_{W \times D})$       ▷ Initialize window with snapshot-dependent rolling noise
5: **while** $|\mathcal{S}| < T_{\text{forecast}}$ **do**       ▷ Predict snapshot $|\mathcal{S}|+1$
6:     $t_{\text{next}} \leftarrow t_{\text{cur}} + \Delta t$       ▷ Global diffusion time after denoising
7:     $\boldsymbol{\sigma}_{\text{cur}} \leftarrow \bar{\boldsymbol{\sigma}}(t_{\text{cur}})$       ▷ Current noise levels
8:     $\boldsymbol{\sigma}_{\text{next}} \leftarrow \bar{\boldsymbol{\sigma}}(t_{\text{next}})$       ▷ Noise levels at the end of this iteration
9:     $\hat{\boldsymbol{y}} \leftarrow D_\theta(\bar{\boldsymbol{x}}_{\text{cur}}, \boldsymbol{\sigma}_{\text{cur}})$       ▷ Denoise sequence
10:     $\boldsymbol{d} \leftarrow (\bar{\boldsymbol{x}}_{\text{cur}} - \hat{\boldsymbol{y}})/\boldsymbol{\sigma}_{\text{cur}}$       ▷ Evaluate $\mathrm{d}\bar{\boldsymbol{x}}/\mathrm{d}t$ at $t_{\text{cur}}$
11:     $\bar{\boldsymbol{x}}_{\text{next}} \leftarrow \bar{\boldsymbol{x}}_{\text{cur}} + (\boldsymbol{\sigma}_{\text{next}} - \boldsymbol{\sigma}_{\text{cur}})\boldsymbol{d}$       ▷ Euler step from $t_{\text{cur}}$ to $t_{\text{next}}$
12:     $n_{\text{clean}} \leftarrow \lfloor t_{\text{next}} \rfloor$       ▷ Infer finished snapshots. If 0, the following lines are a no-op
13:     **add** first $n_{\text{clean}}$ snapshots of $\hat{\boldsymbol{y}}$ to $\mathcal{S}$ and discard first $n_{\text{clean}}$ snapshots of $\bar{\boldsymbol{x}}_{\text{next}}$
14:     **sample** $\boldsymbol{x}_{\text{new}} \sim \mathcal{N}(\mathbf{0}, \sigma_{\text{max}}^2 \mathbf{I}_{n_{\text{clean}} \times D})$
15:     $\bar{\boldsymbol{x}}_{\text{cur}} \leftarrow [\bar{\boldsymbol{x}}_{\text{next}}, \boldsymbol{x}_{\text{new}}]$       ▷ Concatenate fresh noisy snapshots to futuremost
16:     $t_{\text{cur}} \leftarrow t_{\text{next}} - n_{\text{clean}}$       ▷ Re-adjust global diffusion 'time' to be in $[0, 1)$
17: **return** $\mathcal{S}$

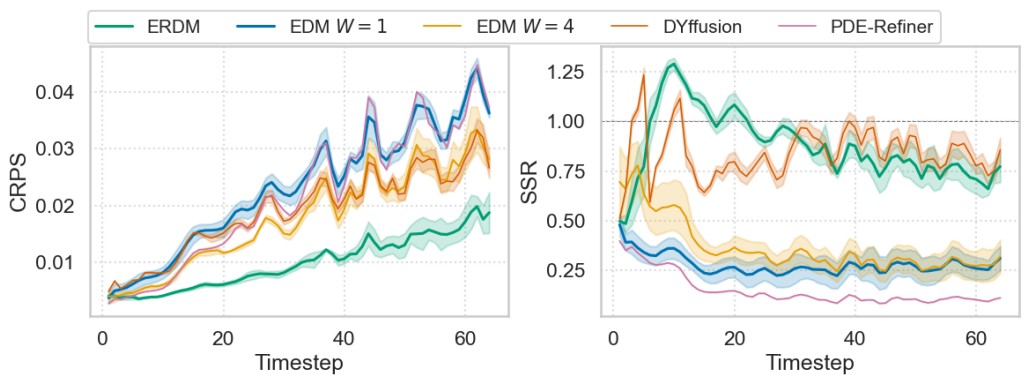

Figure 4: Navier-Stokes test rollout over 64 time steps with 50 ensemble members. ERDM superior performance in both CRPS and calibration compared to single- and multi-step EDM baselines, except for the initial 3 timesteps. Beyond timestep 15, ERDM consistently delivers an approximate 50% improvement in CRPS over the next best model (EDM $W = 4$), demonstrating particular strength in long-range forecasting scenarios.

## 5.2 Navier-Stokes fluid dynamics

**Dataset.** We use the Navier-Stokes fluid dynamics benchmark from [41], defined on a $221 \times 42$ grid. Each simulation features four randomly placed circular obstacles influencing the flow, with fluid viscosity set to $1 \times 10^{-3}$. The dataset comprises $x$ and $y$ velocities and pressure fields. All models receive boundary conditions and obstacle masks as auxiliary inputs. For testing, models predict a 64-timestep trajectory from a single initial snapshot.

**Baselines.** We benchmark ERDM against DYffusion [50], the current state-of-the-art method on this dataset, and PDE-Refiner [32]. To ensure a fair comparison, we retrained DYffusion using our experimental setup, achieving an improvement of over $3\times$ on its originally reported CRPS scores. Our primary focus, however, is to evaluate ERDM's performance relative to common EDM-derived approaches for dynamics forecasting. Consequently, our key baseline is an EDM denoiser parameterized by $D_\theta(\boldsymbol{x}_1 \ldots, \boldsymbol{x}_W; \sigma, \boldsymbol{y}_0)$, where all $\boldsymbol{x}_w$ are corrupted according to the same $\sigma$. With $W = 1$ we recover a next-step forecasting conditional EDM as in [45, 29]. We tried $W \in \{1, 2, 4, 6\}$, and found $W = 4$ to work best for EDM. All baselines are trained on three random seeds and share the same architecture as much as possible. For ERDM, we use $W = 6, \sigma_{\text{min}} = 0.002, \sigma_{\text{max}} = 200, \rho = -10, P_{\text{mean}} = 0.5, P_{\text{std}} = 1.2, N = 1.25$. Further details are provided in Appendix D.2.

**Results.** In Fig. 4, we benchmark ERDM against the baselines on forecasting the five Navier-Stokes test trajectories of length 64. While the EDM-based baselines and PDE-Refiner start well

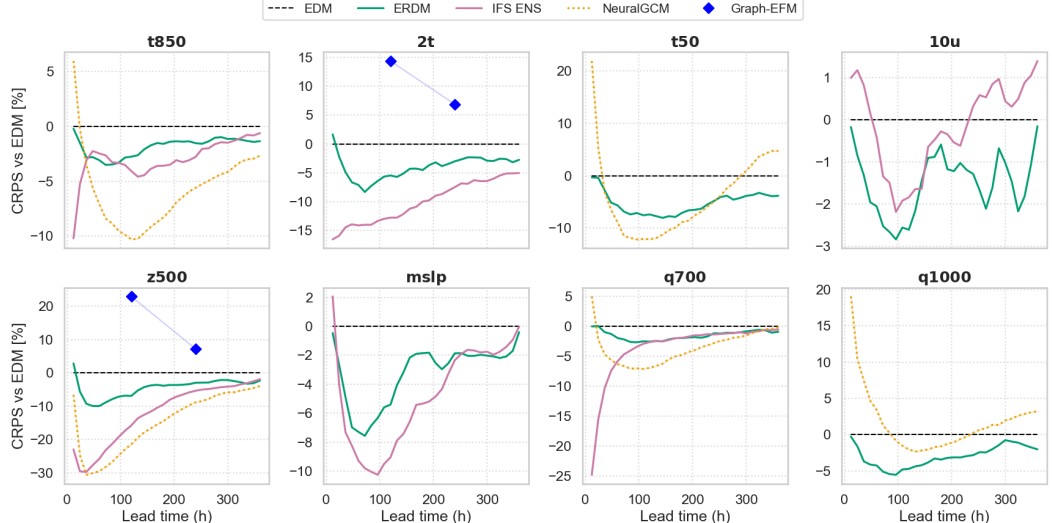

Figure 5: Relative CRPS (lower is better) over single-step EDM baseline as a function of lead time, up to 15 days, for 8 selected variables. ERDM consistently outperforms EDM across most variables and lead times by up to $10\%$. Furthermore, it performs competitively against the state-of-the-art operational physics-based model IFS ENS and the hybrid model NeuralGCM ENS, especially for long lead times, while being more efficient.

in terms of CRPS, even beating ERDM for the very first few time steps, they exhibit a significantly higher error growth than ERDM. We attribute this to the explicit modeling of progressive uncertainty built into our model. At the end of the rollout, ERDM achieves a $50\%$ better CRPS than the best baseline, EDM with a window size of $W = 4$. Similarly, DYffusion starts worse and catches up with the EDM $W = 4$ baseline at the end of the rollout, but not with ERDM. In terms of calibration, we observe that ERDM consistently outperforms the EDM baselines, which are noticeably under-calibrated. We note that this is despite comprehensive tuning of these baselines, where we were only able to achieve calibration improvements at the significant expense of CRPS skill (not shown). Examples of generated forecasts are visualized as videos at this URL: https://youtu.be/jRcwZw5JLe0.

## 5.3 ERA5 weather forecasting

**Dataset.** We benchmark our model on medium-range weather forecasting on the ERA5 reanalysis dataset. We use the $1.5°$ resolution of the data (a $240 \times 121$ grid) provided by [46], with 69 prognostic (input and output) variables: Temperature (t), geopotential (z), specific humidity (q), and the u and v components of wind (u,v) over 13 pressure levels (numbers after the abbreviation refer to the level in hPa) as well as the four surface variables 2m temperature (2t), mean sea level pressure (mslp), and u and v components of winds at 10m (10u, 10v). We train our models on 12-hourly data, as in [45], from 1979 to 2020 and evaluate on 64 2021 initial conditions at 00/12 UTC.

**Baselines.** Our primary baseline is a conditional next-step forecasting EDM, trained using a methodology consistent with ERDM's development. To gauge the absolute performance of ERDM, we also benchmark it against two prominent external models: (1) IFS ENS: The European Centre for Medium-Range Weather Forecasts' (ECMWF) operational, physics-based ensemble forecasting system [15]; (2) NeuralGCM ENS: A hybrid ML-physics stochastic model [28]. Official $1.5°$-resolution forecasts for both external models were sourced from Weatherbench-2 [46] for all variables where available. The IFS ENS evaluation uses the same 2021 initial condition dates as our EDM and ERDM models, but is verified against operational *analyses*.[2] Due to the unavailability of NeuralGCM forecasts for 2021, we use its 2020 forecasts and evaluate them against the corresponding 2020 ERA5 data. Our conditional EDM baseline can be seen as a reproduction of GenCast [45], for which forecasts are only available at higher spatial resolutions and a limited set of variables, but using our experimental setup and neural architecture. Notably, training ERDM is considerably less computationally expensive than NeuralGCM ENS $1.5°$ (GenCast $1°$), which required 128 (32) v5 TPUs and 10 (3.5) days to train. In contrast,

---

[2]As opposed to ERA5 targets. This choice is standard practice [46] and favorable to IFS ENS.

ERDM was trained on 4 H200 GPUs in 5 days only. Lastly, we also include scores for `z500` and `2t` from Graph-EFM [40], taken from their paper. These scores are provided for reference, noting that Graph-EFM is trained on 20 more years and evaluated in 2020. For ERDM, we use $W = 6, \sigma_{\min} = 0.002, \sigma_{\max} = 500, \rho = -10, P_{\mean} = 2, P_{\std} = 1.2, N = 2$. See Appendix D.2 for more experimental details.

**Results.** An evaluation of ERDM's performance, illustrated in Fig. 5, reveals its consistent superiority in relative CRPS skill over our primary EDM baseline, by up to $10\%$. Similarly, ERDM demonstrates a clear advantage over related methods, significantly outperforming Graph-EFM on all four CRPS metrics reported in their study [40]. Against more computationally demanding models such as IFS ENS and NeuralGCM, our approach is broadly competitive. Its primary limitation is a noted weakness in some short-range forecasts compared to IFS ENS, which we attribute to our EDM-based initialization strategy and a backbone architecture not fully specialized for weather prediction. In terms of spread-skill ratio, ERDM, along with IFS ENS, delivers the most calibrated forecasts (see Fig. 12). In contrast, EDM and NeuralGCM often produce underdispersed short-range forecasts. A crucial aspect of evaluating any ensemble forecast is its physical realism. In this regard, ERDM performs exceptionally well. As depicted in Fig. 6, its normalized power spectra are on par with those of the physics-based IFS ENS. This is a significant achievement, as many ML models typically struggle to reproduce such physically consistent spectra—a challenge evident with NeuralGCM in Fig. 6 and documented by Rasp et al. [46]. See Appendix F for extended results, including analyses of extra variables, spread-skill ratios, visualizations, and an EDM vs. ERDM score card.

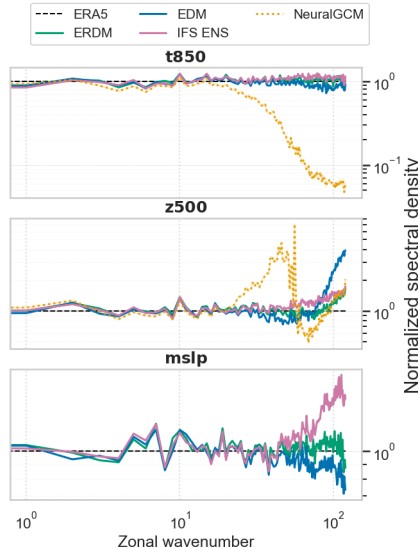

Figure 6: Normalized spectral density of 14-day forecasts, averaged over high latitudes, $[60°, 90°]$. Spectra are divided by the target ERA5 reanalysis spectra. ERDM generates highly accurate spectra that match or slightly beat the physics-based model IFS ENS. NeuralGCM ENS underestimates energy at the mid to high frequencies. See Appendix F.3 for more variables and absolute spectra.

### 5.4 Ablations

We conducted a comprehensive ablation study on the Navier-Stokes task to validate ERDM's design. Our findings reveal that several architectural and training strategies are fundamental to its success; their removal causes performance to collapse, often falling below that of the single-step EDM baseline. In contrast, the model is robust to certain hyperparameters, showing little sensitivity to the window size, $W$, or the noise bounds, $\sigma_{\min}$ and $\sigma_{\max}$, within a reasonable range. We focus on the most impactful ablations below and provide a full analysis in Appendix E.

1. **Appropriate progressive noise schedule & fixed training:** The design of the noise schedule is paramount. Default EDM hyperparameters ($\rho = 7$) prove suboptimal for ERDM, yielding $2\times$ worse CRPS than our proposed default schedule ($\rho = -10$). Performance is robust within a reasonable range of $\rho \in [-30, -5]$, see Fig. 2 for intuition on why. Furthermore, training with a fixed, optimized noise schedule is critical; randomizing the schedule during training degrades performance by nearly $2\times$. Despite this, such randomized training can serve as a heuristic to identify an effective fixed schedule for subsequent retraining. This finding contrasts with the randomized training procedure from Chen et al. [9].

2. **Strategic loss weighting:** Reweighing the losses based on the lognormal probability density function of noise levels, $f(\sigma)$, as proposed, is crucial. Removing this weighting causes a $> 2\times$ performance drop. We generally found that a larger mean of the lognormal distribution, $P_{\mean} > 0$, than EDM's default, $-1.2$, was necessary for ERDM to achieve optimal results, confirming the importance of tuning here.

3. **Dedicated spatiotemporal architecture:** A crucial element is the use of a proper architecture that explicitly models spatiotemporal dependencies. Naively stacking the time di-

mension into the channel dimension of a 2D architecture results in severe $4\times$ performance degradation, confirming the need for our bespoke temporal architecture.

## 5.5 Computational complexity

To analyze the computational demands of ERDM relative to a standard autoregressive EDM, we benchmarked key efficiency metrics on a single A100 GPU. The results, summarized in Table 1, correspond to generating a 30-step (15-day), 5-member ensemble weather forecast. The primary trade-off lies in the architectural design. ERDM's hybrid 3D denoiser is inherently more memory-intensive than the 2D architecture used by EDM, requiring more than twice as much GPU memory. However, the rolling window mechanism makes ERDM significantly more efficient in terms of Neural Function Evaluations (NFEs). ERDM requires $5\times$ fewer NFEs to generate the full 15-day forecast than the step-by-step EDM. Thus, despite the higher cost per step, ERDM's total inference time (including the initialization cost) is competitive with, and even slightly faster than, EDM's.

Table 1: Efficiency metrics for ERDM vs. EDM. All measurements were performed on a single A100 GPU using mixed precision. Inference figures correspond to a 15-day (30 time steps), 5-member ensemble weather forecast with second-order Heun solver. Training uses a batch size of 1. NFEs are Neural Function Evaluations.

| Model | Inference NFEs | Inference Time (s) | GPU Memory (GB) | |
| --- | --- | --- | --- | --- |
| | | | Inference | Training |
| EDM | 600 | 237 | 21 | 19 |
| ERDM | 120 | 209 | 49 | 53 |

## 6 Conclusion

We introduced the Elucidated Rolling Diffusion Model (ERDM), a diffusion framework for long-range probabilistic forecasting in complex scientific systems. ERDM adapts EDM diffusion to sequential data by integrating a progressive temporal noise schedule and snapshot-dependent preconditioning, enabling it to explicitly model the increasing uncertainty of chaotic dynamics. Our ablation studies validate that these components, combined with a bespoke spatiotemporal architecture, are synergistic and essential for strong performance. On challenging benchmarks like ERA5 weather data and Navier-Stokes simulations, ERDM consistently outperforms relevant baselines in probabilistic metrics such as CRPS, calibration, and physical realism–particularly over extended forecast horizons. Future work could improve computational efficiency through latent-space modeling or explore the framework's applicability to a wider range of physical systems.

**Limitations.** The primary limitation of ERDM is the computational cost of its 3D denoiser architecture. While ERDM requires fewer sampling steps than autoregressive baselines (e.g., $40\%$ less on ERA5), each step is more expensive, resulting in a comparable total inference time but significantly higher memory usage. This memory complexity poses a key barrier to scaling to higher resolutions, though techniques like gradient checkpointing or latent diffusion could offer a path forward. Beyond computational hurdles, ERDM's short-range weather forecasts are not yet on par with leading operational models like IFS ENS. The framework is also constrained by its reliance on an external model for initialization and its use of an explicit noise-level loss weighting, which may be suboptimal compared to importance sampling [14], offering exciting directions for future work.

## Acknowledgements

S.R.C. acknowledges generous support from the summer internship and collaboration at NVIDIA. This research used resources from the National Energy Research Scientific Computing Center (NERSC), a Department of Energy User Facility, using NERSC awards DDR-ERCAP0034142, ASCR-ERCAP0033209, and EESSD-ERCAP0033799. This work was supported in part by the U.S. Army Research Office under Army-ECASE award W911NF-07-R-0003-03, the U.S. Department Of Energy, Office of Science, IARPA HAYSTAC Program, and NSF Grants #2205093, #2146343, #2134274, CDC-RFA-FT-23-0069, DARPA AIE FoundSci and DARPA YFA. We are grateful to the anonymous reviewers for their valuable feedback that helped strengthen this work.

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

# Appendix

# Contents

## A  Broader Impact

Our research holds the potential for societal benefits, primarily through improved disaster preparedness, enhanced decision-making across sectors like agriculture or energy. However, these advancements also necessitate careful consideration of potential negative impacts, such as over-reliance on ML-based forecasts or socioeconomic disparities in access and benefit. Mitigating these risks requires comprehensive evaluations, communication about forecast uncertainties, and efforts to ensure equitable access to these new tools. Any deployment of our model, or derived versions, should be accompanied by either more comprehensive evaluations than in our paper, or careful disclaimers to not blindly trust its forecasts, especially for data outside of the training data distribution.

## B  Background on Diffusion Models

Consider the data distribution represented by $p_{\text{data}}(\boldsymbol{x})$. The forward diffusion process transforms this distribution by incorporating time-dependent Gaussian noise, yielding modified distributions

$p(\boldsymbol{x}; \mathbf{G}(t))$. This transformation is achieved by adding Gaussian noise with zero mean and covariance $\mathbf{G}(t)$ to the data. $\mathbf{G}(t)$ is a $d \times d$ positive definite noise covariance matrix that evolves over diffusion time $t$. When the "magnitude" of $\mathbf{G}(t)$ (e.g., its trace or smallest eigenvalue) is sufficiently large, the resulting distribution $p(\boldsymbol{x}; \mathbf{G}(t))$ approximates a pure Gaussian distribution $\mathcal{N}(\mathbf{0}, \mathbf{G}(t))$.

Conversely, the backward diffusion process operates by initially sampling noise, represented as $\boldsymbol{x}_0$, from a prior distribution, typically $\mathcal{N}(\mathbf{0}, \mathbf{G}(0))$, where $\mathbf{G}(0)$ corresponds to the maximum noise level. The process then focuses on denoising this sample through a sequence, $\boldsymbol{x}_i$, characterized by a sequence of decreasing noise covariance matrices: $\mathbf{G}(t_0) > \mathbf{G}(t_1) > \ldots > \mathbf{G}(t_N) \approx \mathbf{0}$ (in the Loewner order), where $t_0 = 0$. Each intermediate sample $\boldsymbol{x}_i$ is drawn from $\boldsymbol{x}_i \sim p(\boldsymbol{x}_i; \mathbf{G}(t_i))$. The terminal sample of the backward process, $\boldsymbol{x}_N$, is expected to approximate the original data distribution $p_{\text{data}}(\boldsymbol{y})$. Note that for the sake of consistency with our ERDM formulation, we reversed the diffusion time order. That is, $t = 0$ (1) corresponds to maximal (minimal) noise levels, which is sometimes described in opposite order in the diffusion literature.

**SDE formulation**. To present the forward and backward processes rigorously, they can be captured via stochastic differential equations (SDEs). Such SDEs ensure that the sample, $\boldsymbol{x}(t)$, aligns with the designated data distribution, $p(\boldsymbol{x}; \mathbf{G}(t))$, as it evolves through diffusion time $t$ [54]. The Elucidated Diffusion Model (EDM) by Karras et al. [26] provides a principled design for diffusion models, typically based on a scalar noise schedule $\sigma(t)$ (leading to isotropic noise $\sigma^2(t)\mathbf{I}$). The formulation presented here extends these principles to accommodate a general, potentially anisotropic, time-varying noise covariance matrix $\mathbf{G}(t)$. Let $\dot{\mathbf{G}}(t) = d\mathbf{G}(t)/dt$ be the time derivative of the noise covariance matrix. We assume $\dot{\mathbf{G}}(t)$ is positive semi-definite for all $t$, and let $\mathbf{K}(t)$ be any matrix such that $\mathbf{K}(t)\mathbf{K}(t)^T = \dot{\mathbf{G}}(t)$ (e.g., via Cholesky decomposition if $\dot{\mathbf{G}}(t)$ is strictly positive definite, or other matrix square roots). The forward SDE, describing the process of adding noise such that $\boldsymbol{x}(t) - \boldsymbol{x}(1) \sim \mathcal{N}(\mathbf{0}, \mathbf{G}(t))$ (assuming $\mathbf{G}(1) = \mathbf{0}$, and $\boldsymbol{x}(1) \sim p_{\text{data}}(\boldsymbol{y})$), is:

$$d\boldsymbol{x} = \mathbf{K}(t)d\boldsymbol{\omega}(t), \tag{5}$$

where $\boldsymbol{\omega}(t)$ is a standard $d$-dimensional Wiener process. The corresponding backward (reverse-time) SDE is given by [54]:

$$d\boldsymbol{x} = -\dot{\mathbf{G}}(t)\nabla_{\boldsymbol{x}} \log p(\boldsymbol{x}; \mathbf{G}(t))dt + \mathbf{K}(t)d\bar{\boldsymbol{\omega}}(t), \tag{6}$$

where $\bar{\boldsymbol{\omega}}(t)$ is a standard Wiener process in reverse time. The term 'time' $t$ here is a conceptual dimension for the denoising steps. The backward SDE comprises a deterministic drift term related to the score of the perturbed data distribution and a stochastic noise injection term.

**Denoising score matching**. An examination of the SDE in equation 6 indicates the necessity of the score function, $\nabla_{\boldsymbol{x}} \log p(\boldsymbol{x}; \mathbf{G}(t))$, for sampling. This score function can be estimated using the relationship derived from Tweedie's formula. Given that $\boldsymbol{x} = \boldsymbol{y} + \boldsymbol{\epsilon}$ where $\boldsymbol{y} \sim p_{\text{data}}(\boldsymbol{y})$ and $\boldsymbol{\epsilon} \sim \mathcal{N}(\mathbf{0}, \mathbf{G}(t))$, the score is:

$$\nabla_{\boldsymbol{x}} \log p(\boldsymbol{x}; \mathbf{G}(t)) = (\mathbf{G}(t))^{-1}(\mathbb{E}[\boldsymbol{y}|\boldsymbol{x}, \mathbf{G}(t)] - \boldsymbol{x}). \tag{7}$$

A denoising neural network, $D_\theta(\boldsymbol{x}; \mathbf{G}(t))$, is trained to approximate the conditional expectation $\mathbb{E}[\boldsymbol{y}|\boldsymbol{x}, \mathbf{G}(t)] \approx \boldsymbol{y}$. Thus, the score is approximated as $(\mathbf{G}(t))^{-1}(D_\theta(\boldsymbol{x}; \mathbf{G}(t)) - \boldsymbol{x})$. The network is trained by minimizing the following objective, typically by sampling $t$ (and thus $\mathbf{G}(t)$) according to a predefined schedule or distribution $p_t(t)$ over diffusion times:

$$\min_\theta \mathbb{E}_{\boldsymbol{y} \sim p_{\text{data}}} \mathbb{E}_{t \sim p_t(t)} \mathbb{E}_{\boldsymbol{\epsilon} \sim \mathcal{N}(\mathbf{0}, \mathbf{G}(t))} \left[ \|D_\theta(\boldsymbol{y} + \boldsymbol{\epsilon}; \mathbf{G}(t)) - \boldsymbol{y}\|_2^2 \right]. \tag{8}$$

The denoiser $D_\theta$ is now conditioned on the full noise covariance matrix $\mathbf{G}(t)$.

**Sampling**. To generate samples from the model, one typically discretizes the backward SDE equation 6 and simulates it from $t = 0$ to $t \approx 1$. A common discretization is the Euler-Maruyama method. Given a sequence of diffusion times $0 = \tau_0 < \tau_1 < \ldots < \tau_N \approx 1$, and an initial sample $\boldsymbol{x}_{\tau_0} \sim \mathcal{N}(\mathbf{0}, \mathbf{G}(0))$, the update rule for $i = 0, \ldots, N-1$ is:

$$\boldsymbol{x}_{\tau_{i+1}} = \boldsymbol{x}_{\tau_i} - (\mathbf{G}(\tau_i) - \mathbf{G}(\tau_{i+1})) \hat{\nabla}_{\boldsymbol{x}} \log p(\boldsymbol{x}_{\tau_i}; \mathbf{G}(\tau_i)) + \sqrt{\mathbf{G}(\tau_i) - \mathbf{G}(\tau_{i+1})} \mathbf{z}_i, \tag{9}$$

where $\Delta\mathbf{G}(\tau_i) = \mathbf{G}(\tau_i) - \mathbf{G}(\tau_{i+1}) > \mathbf{0}$, we have used the approximation $\dot{\mathbf{G}}(\tau_i)\Delta\tau_i \approx \mathbf{G}(\tau_i) - \mathbf{G}(\tau_{i+1})$ (assuming $\Delta\mathbf{G}(\tau_i)$ is small), and $\mathbf{z}_i \sim \mathcal{N}(\mathbf{0}, \mathbf{I})$ is a standard Gaussian noise sample. The term $\hat{\nabla}_{\boldsymbol{x}} \log p(\boldsymbol{x}_{\tau_i}; \mathbf{G}(\tau_i))$ is the score estimated using the trained denoiser:

$$\hat{\nabla}_{\boldsymbol{x}} \log p(\boldsymbol{x}_{\tau_i}; \mathbf{G}(\tau_i)) = (\mathbf{G}(\tau_i))^{-1}(D_\theta(\boldsymbol{x}_{\tau_i}; \mathbf{G}(\tau_i)) - \boldsymbol{x}_{\tau_i}). \tag{10}$$

More sophisticated samplers, such as those proposed by Karras et al. [26] (e.g., second-order Heun or DPM-Solver++), can also be adapted for the general covariance case to improve sample quality or reduce the number of sampling steps $N$. These often involve more complex update rules, potentially incorporating predictor-corrector steps or higher-order approximations of the SDE.

**EDM preconditioning.** Recall the denoiser network parameterization in EDM:

$$D_\theta(\boldsymbol{x}; \sigma) = c_{\text{skip}}(\sigma)\boldsymbol{x} + c_{\text{out}}(\sigma)F_\theta(c_{\text{in}}(\sigma)\boldsymbol{x}, c_{\text{noise}}(\sigma)), \tag{11}$$

where $\boldsymbol{x}$ corresponds to data corrupted with Gaussian noise of standard deviation $\sigma$ and $F_\theta$ is the raw neural network trained to denoise $\boldsymbol{x}$. One of the key contributions in EDM is the careful choice of preconditioning ($c_{\text{skip}}, c_{\text{out}}, c_{\text{in}}$). The preconditioning ensures unit variance for the denoiser network's inputs and targets. Consistent scales, regardless of noise levels, simplify the task for the denoiser network, which does not need to adapt to changing magnitudes for different noise standard deviations. Mathematically, these functions are defined as:

$$c_{\text{in}}(\sigma) = \frac{1}{\sqrt{\sigma^2 + \sigma_{\text{data}}^2}} \quad c_{\text{skip}}(\sigma) = \frac{\sigma_{\text{data}}^2}{\sigma^2 + \sigma_{\text{data}}^2} \quad c_{\text{out}}(\sigma) = \frac{\sigma \cdot \sigma_{\text{data}}}{\sqrt{\sigma^2 + \sigma_{\text{data}}^2}} \tag{12}$$

The transformation of the noise level is chosen empirically as $c_{\text{noise}}(\sigma) = \ln(\sigma)/4$. In ERDM, we vectorize the preconditioning to function on each snapshot independently, thus ensuring that the same EDM design principles apply to the noisy sequence that ERDM's denoiser network ingests.

## C  ERDM

### C.1  Probability flow ODE derivation

Recall that ERDM operated on noisy sequences $\bar{\boldsymbol{x}}_{1:W}$ of size $W$, with each $\bar{\boldsymbol{x}}_w \in \mathbb{R}^D$ corrupted according to increasing noise level standard deviations $0 \approx \bar{\sigma}_1(t) < \bar{\sigma}_2(t) < \cdots < \bar{\sigma}_W(t)$. The noise covariance matrix for the entire window $\bar{\boldsymbol{x}}_{1:W}$ at diffusion time $t$ is block-diagonal:

$$\mathbf{G}_{\text{window}}(t) = \text{diag}(\bar{\sigma}_1^2(t)\mathbf{I}_D, \bar{\sigma}_2^2(t)\mathbf{I}_D, \ldots, \bar{\sigma}_W^2(t)\mathbf{I}_D). \tag{13}$$

Its time derivative is:

$$\dot{\mathbf{G}}_{\text{window}}(t) = \text{diag}(2\bar{\sigma}_1(t)\dot{\sigma}_1(t)\mathbf{I}_D, \ldots, 2\bar{\sigma}_W(t)\dot{\sigma}_W(t)\mathbf{I}_D). \tag{14}$$

Let $\mathbf{K}_{\text{window}}(t)$ be a matrix such that $\mathbf{K}_{\text{window}}(t)\mathbf{K}_{\text{window}}(t)^T = \dot{\mathbf{G}}_{\text{window}}(t)$. Specifically:

$$\mathbf{K}_{\text{window}}(t) = \text{diag}(\sqrt{2\bar{\sigma}_1(t)\dot{\sigma}_1(t)}\mathbf{I}_D, \ldots, \sqrt{2\bar{\sigma}_W(t)\dot{\sigma}_W(t)}\mathbf{I}_D), \tag{15}$$

assuming $\bar{\sigma}_w(t)\dot{\sigma}_w(t) \geqslant 0$. The forward SDE for the window $\boldsymbol{x}(t)$ is:

$$\mathrm{d}\boldsymbol{x} = \mathbf{K}_{\text{window}}(t)\mathrm{d}\boldsymbol{\omega}(t), \tag{16}$$

where $\boldsymbol{\omega}(t)$ is a standard $WD$-dimensional Wiener process. The corresponding backward (reverse-time) SDE, which is the primary SDE used for generation, is [54]:

$$\mathrm{d}\bar{\boldsymbol{x}} = \left[-\dot{\mathbf{G}}_{\text{window}}(t)\nabla_{\bar{\boldsymbol{x}}} \log p(\bar{\boldsymbol{x}}; \mathbf{G}_{\text{window}}(t))\right]\mathrm{d}t + \mathbf{K}_{\text{window}}(t)\mathrm{d}\bar{\boldsymbol{\omega}}(t). \tag{17}$$

The score for the $w$-th snapshot of the window, $\bar{\boldsymbol{x}}_w$, is approximated with the learned denoiser, $D_\theta$:

$$\nabla_{\bar{\boldsymbol{x}}_w} \log p_\theta(\bar{\boldsymbol{x}}; \mathbf{G}_{\text{window}}(t)) = (\bar{\sigma}_w^2(t))^{-1}(D_\theta(\bar{\boldsymbol{x}}; \bar{\boldsymbol{\sigma}}(t))_w - \bar{\boldsymbol{x}}_w), \tag{18}$$

where $D_\theta(\bar{\boldsymbol{x}}; \bar{\boldsymbol{\sigma}}(t))_w$ is the $w$-th snapshot output of the denoiser $D_\theta$, which is conditioned on the full noisy window $\bar{\boldsymbol{x}}$ and the vector of current noise standard deviations $\bar{\boldsymbol{\sigma}}(t)$. The probability flow ODE, which provides a deterministic path for generation, is obtained by removing the stochastic term from the backward SDE and adjusting the drift (see Song et al. [54], Eq. (14) and Karras et al. [26], Eq. (5)):

$$\mathrm{d}\bar{\boldsymbol{x}} = \left[-\dot{\mathbf{G}}_{\text{window}}(t)\nabla_{\bar{\boldsymbol{x}}} \log p_\theta(\bar{\boldsymbol{x}}; \mathbf{G}_{\text{window}}(t))\right]\mathrm{d}t. \tag{19}$$

Using the specific EDM parameterization where the ODE drift is $-\mathbf{D}(t)\nabla_{\bar{\boldsymbol{x}}} \log p(\bar{\boldsymbol{x}}; \bar{\boldsymbol{\sigma}}(t))$ with $\mathbf{D}(t) = \text{diag}(\bar{\sigma}_1(t)\dot{\sigma}_1(t)\mathbf{I}_D, \ldots, \bar{\sigma}_W(t)\dot{\sigma}_W(t)\mathbf{I}_D)$, the ODE becomes:

$$\mathrm{d}\bar{\boldsymbol{x}} = -\text{diag}(\bar{\sigma}_1(t)\dot{\sigma}_1(t)\mathbf{I}_D, \ldots, \bar{\sigma}_W(t)\dot{\sigma}_W(t)\mathbf{I}_D)\nabla_{\bar{\boldsymbol{x}}} \log p_\theta(\bar{\boldsymbol{x}}; \bar{\boldsymbol{\sigma}}(t))\mathrm{d}t. \tag{20}$$

## C.2 Sampling

ERDM generates sequences with a sliding window of size $W$. Generating each window requires solving the backward SDE in equation 17 (or ODE in equation 20). In each iteration $k$ where the SDE is fully solved, ERDM operates on the noisy window $\bar{x}_{1:W}^{(k)}$. After solving the SDE backwards $k \geqslant 1$ times, the first snapshot of the window, $\bar{x}_1^{(k)}$ (with relative window index $w = 1$), corresponds to a forecast of $y_k$, where $k$ is the *absolute* time step of the forecast time dimension. Solving the SDE requires discretizing it into $N_k$ diffusion "time" steps, each of size $\Delta t_k := \frac{1}{N_k}$. In general, $N_k$ may vary across iterations. To simplify our illustrations of the sampling process in the following, we assume that $N_k$ and $\Delta t_k$ are identical for all $k$, denoted as $N$ and $\Delta t$ respectively. This can be achieved by setting $N$ in our sampling algorithms to a positive integer. Note that in practice $N$ need not be an integer (e.g., we use $N = 1.25$ for our Navier-Stokes experiments), which is accounted for in our sampling algorithms. With this simplification, the SDE evolves, for all $k$, following $\bar{\sigma}(t_1) \to \bar{\sigma}(t_2) \to \cdots \to \bar{\sigma}(t_N)$, where $t_{i+1} = t_i + \Delta t$ for $i \geqslant 1$, $t_1 = 0$, and $t_N = 1$. For example, if $N = 1$ the SDE is solved from $\bar{\sigma}(0)$ to $\bar{\sigma}(1)$ in one sampling step, while if $N = 2$ it would be solved following $\bar{\sigma}(0) \to \bar{\sigma}(\frac{1}{2}) \to \bar{\sigma}(1)$ in two sampling steps (see illustration in Fig. 1). At the end of the $N$ sampling steps, $\bar{\sigma}_1(1) = \sigma_{\min}$ and the first snapshot in the window can be emitted as the forecast for lead time $k$, $\bar{x}_1^{(k)} \approx y_k$. In the next iteration of $N$ sampling steps, the next snapshot $k + 1$ will be predicted by operating on a new window, $\bar{x}_{1:W}^{(k+1)}$. Because $\bar{\sigma}_{w+1}(1) = \bar{\sigma}_w(0)$ for $w \geqslant 1$, this new window can be constructed by shifting the noisy parts of the old window by one position. That is, $\bar{x}_w^{(k+1)} := \bar{x}_{w+1}^{(k)}$ for $w < W$, and a new pure-noise snapshot is appended as $\bar{x}_W^{(k+1)} \sim \mathcal{N}(\mathbf{0}, \sigma_{\max}^2 \mathbf{I})$. The new window is thus noised according to standard deviations $\bar{\sigma}(0)$ and the process can be restarted and repeated indefinitely, as illustrated in Fig. 1.

**Discretization.** As mentioned above, the SDE or probability flow ODE need to be solved through discretization over $N$ diffusion steps. We denote the noisy window at diffusion step $n \in \{1, \ldots, N\}$ as $\bar{x}(n)$. We ignore the superscript $k$ since the ODE discretization is independent of it. Assuming that $\bar{x}(0)$ corresponds to a window corrupted according to $\bar{\sigma}(0)$, an Euler–Maruyama step for the probability flow ODE and the whole window is

$$\bar{x}(n + 1) = \bar{x}(n) + \frac{\bar{\sigma}((n + 1)\Delta t) - \bar{\sigma}(n\Delta t)}{\bar{\sigma}(n\Delta t)} \big[\bar{x}(n) - D_\theta\big(\bar{x}(n); \bar{\sigma}(n\Delta t)\big)\big] \qquad (21)$$

The ODE is thus solved *in parallel* for all snapshots in the window. When $\bar{x}(N)$ is reached, its noise levels correspond to $\bar{\sigma}(N\Delta t) = \bar{\sigma}(1)$. Thus, the first snapshot is emitted, and the window is shifted as describe above. Higher-order stochastic integrators—e.g. the second-order Heun scheme of EDM [26]—can replace Euler–Maruyama as decribed in the next subsection C.3 below.

**Continuous-time view of a tracked snapshot.** Focus on a single physical snapshot $z$ as it is refined across windows. Let $k$ be the rollout step at which $z$ is finalised. The snapshot first appears at rollout step $k - W + 1$ (relative position $w = W$) and moves to $w = 1$ at step $k$. Define $s_w := \bar{\sigma}_w(0)$ as the initial noise level when the snapshot is at position $w$, and treat $w$ as a continuous variable on the normalised interval $[0, 1]$ (with $w \searrow 0$ as refinement proceeds). Let $z_w$ denote the state of the tracked snapshot when its current noise level is $s_w$. Its evolution is governed by

$$d z_w = -\frac{1}{2} \frac{d s_w^2}{d w} \nabla_{z_w} \log p\big(z_w; s_w, \mathcal{C}_w\big) \, dw + \sqrt{\frac{d s_w^2}{d w}} \, d\bar{\epsilon}_w, \qquad (22)$$

where $dw < 0$ since $w$ decreases. The context $\mathcal{C}_w$ comprises all other snapshots in the window: The past $[0, w)$ and the future $(w, 1]$. Note that $\frac{d s_w^2}{d w} > 0$, and the score is obtained as

$$\nabla_{z_w} \log p\big(z_w; s_w, \mathcal{C}_w\big) = \frac{1}{s_w^2} \big(D_\theta(z_w; s_{1:W}, \mathcal{C}_w)_w - z_w\big),$$

where $D_\theta(z_w; s_{1:W}, \mathcal{C}_w)_w$ is the fully denoised estimate of the snapshot at position $w$. SDE in equation 22 therefore captures how a snapshot is progressively refined as it shifts forward through successive windows, with its effective noise level $s_w$ decreasing monotonically.

## C.3 Stochastic Heun Sampler

In the main text, we introduced our proposed deterministic, first-order sampling algorithm (Algorithm 2). We now extend this into a more sophisticated sampler, detailed in Algorithm 3, by incorporating two key modifications: a second-order Heun correction step and optional sampling stochasticity via a "churn" mechanism. Both components are inspired by EDM's stochastic sampling algorithm [26], but require specific adaptations for our ERDM to work.

**Heun correction step.** The Heun method enhances sampling accuracy by incorporating a corrector step over the basic Euler step during the denoising of the data. However, a complication arises because when $t'_{\text{next}} \geqslant 1$, the first snapshot of $\bar{x}'_{\text{next}}$ is already fully denoised (i.e., "clean"). Applying the denoiser to such a clean snapshot as part of the Heun step would be inappropriate. To address this, we pad the noisy data by one extra pure-noise snapshot to the end of the sequence, resulting in a window of size $W + 1$. The core denoiser, $D_\theta$, is then selectively applied on the sub-window of size $W$, depending on whether the first snapshot is already clean or not. That is, if $t'_{\text{next}} \geqslant 1$, the denoiser would be applied on the snapshots with indices $2, \ldots, W + 1$. Otherwise, it is applied, as usual, to the indices $1, \ldots, W$. To formalize this selective application, we define $D_\theta^{\text{pad}}(\boldsymbol{x}, \bar{\boldsymbol{\sigma}}(t))$ such that it applies the actual denoiser on all snapshots with indices $I = \{\lfloor t \rfloor + 1, \ldots, \lfloor t \rfloor + W\}$ and leaves all other snapshots untouched. This ensures that the denoiser evaluations within the Heun step are confined to snapshots that are indeed noisy.

$$D_\theta^{\text{pad}}(\boldsymbol{x}, \bar{\boldsymbol{\sigma}}(t))_w := \begin{cases} D_\theta(\boldsymbol{x}_I, \bar{\boldsymbol{\sigma}}(t)_I)_{w-\lfloor t \rfloor} & \text{if } w \in I \\ \boldsymbol{x}_w & \text{otherwise} \end{cases} \tag{23}$$

**Churn mechanism adaptation.** The second modification is the integration of a "churn" step, which introduces new noise at each iteration to inject stochasticity into the sampling process. Unlike the EDM algorithm, which applies churn at the beginning of each iteration, we position it at the end of each iteration. This change is motivated by the fact that the first snapshots of the very first initial window are noised based on low levels of noise, so that artificially injecting more noise before performing any denoising could be harmful. This contrasts with the EDM setup, where the noisy data at the very first iteration is typically maximally noisy.

**Experimental configuration.** For the experiments reported in this work, the stochastic Heun sampler (Algorithm 3) is always used with the Heun correction step. Our ablations in Appendix E demonstrate that our second-order sampler provides better results than a first-order version (Euler-only; Algorithm 2). However, we disable sampling stochasticity by setting the churn rate $S_{\text{churn}} = 0$. In our experience, we found that small values of $S_{\text{churn}} \approx 0.1$ can increase the ensemble spread, albeit it had minimal impact on error metrics such as the CRPS. We defer a more comprehensive analysis of stochastic sampling to future work. Our algorithm can, in principle, be used to fully denoise multiple snapshots at once (e.g., output two snapshots per iteration when $N = \frac{1}{2}$). This could be useful in certain modalities or problems where subsequent snapshots are highly correlated, but is left to future work too.

## C.4 Temporal noise prior

In our training and sampling algorithms (Alg. 1-3), we use i.i.d. noise to corrupt the first window as well as append new pure-noise snapshots when sliding the window. As mentioned in the main text, it has been shown to be useful to sample *temporally correlated noise* snapshots. In our experiments, we use the simple "progressive noise model" technique proposed by Ge et al. [18], which we found to improve results on the Navier-Stokes dataset compared to using i.i.d. noise (see Appendix E). This noise prior generates the noise for each snapshot autoregressively by combining a perturbed version of the previous noise with a new i.i.d. noise sample. Mathematically, the noise for the first snapshot is random, $\boldsymbol{\epsilon}^{(1)} \sim \mathcal{N}(\mathbf{0}, \mathbf{I})$, while for the following snapshots $k > 1$:

$$\boldsymbol{\epsilon}^{(k)} = \frac{\alpha}{\sqrt{1 + \alpha^2}} \boldsymbol{\epsilon}^{(k-1)} + \boldsymbol{\epsilon}_{\text{ind}}^{(k)}, \qquad \boldsymbol{\epsilon}_{\text{ind}}^{(k)} \sim \mathcal{N}(\mathbf{0}, \frac{1}{1 + \alpha^2}\mathbf{I}), \tag{24}$$

where $\alpha$ controls the correlation strength between snapshot noises and $\alpha = 0$ corresponds to i.i.d. noise. In our experiments, we use $\alpha = 1$ following the recommended practice in [18]. These temporally correlated noise samples replace the i.i.d. draws in lines 6 and 27 of Algorithm 3. For

---

**Algorithm 3** Elucidated Rolling Diffusion Stochastic Heun Sampler

---

1: **Require:** $\hat{\boldsymbol{y}}_{1:W}^{\text{init}}, N, T_{\text{forecast}}, S_{\text{churn}} = 0, S_{\text{noise}} = 1$      ▷ Choose $S_{\text{churn}} > 0$ for sampling stochasticity
2:    # Initialization
3:    $\Delta t \leftarrow 1/N$      ▷ Infer step size from given number of steps per snapshot
4:    $\Delta t' \leftarrow \Delta t/(1 - S_{\text{churn}})$      ▷ Size of larger, initial step, before backtracking (if $S_{\text{churn}} > 0$) to $\Delta t$
5:    $t_{\text{cur}} \leftarrow 0; \; \mathcal{S} \leftarrow \varnothing$      ▷ Initialize global diffusion time and empty generated sequence
6:    **sample** $\bar{\boldsymbol{x}}_{\text{cur}} \sim \mathcal{N}(\hat{\boldsymbol{y}}_{1:W}^{\text{init}}, \bar{\boldsymbol{\sigma}}(t_{\text{cur}})^2 \mathbf{I}_{W \times D})$      ▷ Initialize window with snapshot-dependent rolling noise
7:    **concatenate** $n_{\text{pad}} := \lfloor \Delta t' \rfloor + 1$ fully noisy snapshots to the end of $\bar{\boldsymbol{x}}_{\text{cur}}$      ▷ Necessary for Heun step
8:    **while** $|\mathcal{S}| < T_{\text{forecast}}$ **do**      ▷ Predict snapshot $|\mathcal{S}|+1$
9:       $t'_{\text{next}} \leftarrow t_{\text{cur}} + \Delta t'$      ▷ Global diffusion time after denoising, before churn
10:      $t_{\text{next}} \leftarrow t_{\text{cur}} + \Delta t$      ▷ Global diffusion time after churn ($t_{\text{next}} \leqslant t'_{\text{next}}$)
11:      $\boldsymbol{\sigma}_{\text{cur}} \leftarrow \bar{\boldsymbol{\sigma}}(t_{\text{cur}})$      ▷ Current noise levels
12:      $\boldsymbol{\sigma}'_{\text{next}} \leftarrow \bar{\boldsymbol{\sigma}}(t'_{\text{next}})$      ▷ Noise levels before churn
13:      $\boldsymbol{\sigma}_{\text{next}} \leftarrow \bar{\boldsymbol{\sigma}}(t_{\text{next}})$      ▷ Noise levels at the end of this iteration
14:      # Euler step
15:      $\hat{\boldsymbol{y}} \leftarrow D_\theta^{\text{pad}}(\bar{\boldsymbol{x}}_{\text{cur}}, \boldsymbol{\sigma}_{\text{cur}})$      ▷ Denoise sequence
16:      $\boldsymbol{d} \leftarrow (\bar{\boldsymbol{x}}_{\text{cur}} - \hat{\boldsymbol{y}})/\boldsymbol{\sigma}_{\text{cur}}$      ▷ Evaluate $\mathrm{d}\boldsymbol{x}/\mathrm{d}\tau$ at $t_{\text{cur}}$
17:      $\bar{\boldsymbol{x}}'_{\text{next}} \leftarrow \bar{\boldsymbol{x}}_{\text{cur}} + (\boldsymbol{\sigma}'_{\text{next}} - \boldsymbol{\sigma}_{\text{cur}})\boldsymbol{d}$      ▷ Euler step from $t_{\text{cur}}$ to $t'_{\text{next}}$
18:      # Heun step (2nd-order correction)
19:      $\boldsymbol{d}' \leftarrow (\bar{\boldsymbol{x}}'_{\text{next}} - D_\theta^{\text{pad}}(\bar{\boldsymbol{x}}'_{\text{next}}, \boldsymbol{\sigma}'_{\text{next}}))/\boldsymbol{\sigma}'_{\text{next}}$      ▷ Denoise at $t'_{\text{next}}$
20:      $\bar{\boldsymbol{x}}'_{\text{next}} \leftarrow \bar{\boldsymbol{x}}_{\text{cur}} + \frac{1}{2}(\boldsymbol{\sigma}'_{\text{next}} - \boldsymbol{\sigma}_{\text{cur}})(\boldsymbol{d} + \boldsymbol{d}')$      ▷ $2^{\text{nd}}$ order Heun correction step
21:      # Churn step (stochastic backtrack)
22:      **sample** $\boldsymbol{\epsilon} \sim \mathcal{N}(\mathbf{0}, S_{\text{noise}}^2 \mathbf{I}_{W \times D})$      ▷ Sample churn noise
23:      $\bar{\boldsymbol{x}}_{\text{next}} \leftarrow \bar{\boldsymbol{x}}'_{\text{next}} + \sqrt{\boldsymbol{\sigma}_{\text{next}}^2 - \boldsymbol{\sigma}_{\text{next}}'^2} \cdot \boldsymbol{\epsilon}$      ▷ Backtrack to $t_{\text{next}}$ noise levels
24:      # Sliding-window shift and forecast snapshot extraction
25:      $n_{\text{clean}} \leftarrow \lfloor t_{\text{next}} \rfloor$      ▷ Infer finished snapshots. If 0, the following lines are a no-op
26:      **add** first $n_{\text{clean}}$ snapshots of $\hat{\boldsymbol{y}}$ to $\mathcal{S}$ and discard first $n_{\text{clean}}$ snapshots of $\bar{\boldsymbol{x}}_{\text{next}}$
27:      **sample** $\boldsymbol{x}_{\text{new}} \sim \mathcal{N}(\mathbf{0}, \sigma_{\text{max}}^2 \mathbf{I}_{n_{\text{clean}} \times D})$
28:      $\bar{\boldsymbol{x}}_{\text{cur}} \leftarrow [\bar{\boldsymbol{x}}_{\text{next}}, \boldsymbol{x}_{\text{new}}]$      ▷ Concatenate fresh noisy snapshots to futuremost
29:      $t_{\text{cur}} \leftarrow t_{\text{next}} - n_{\text{clean}}$      ▷ Re-adjust global diffusion 'time' to be in $[0, 1)$
30: **return** $\mathcal{S}$

---

example, line 6 becomes $\bar{\boldsymbol{x}}_{\text{cur}} \leftarrow \hat{\boldsymbol{y}}_{1:W}^{\text{init}} + \bar{\boldsymbol{\sigma}}(t_{\text{cur}})\boldsymbol{\epsilon}_{1:W}$, where $\boldsymbol{\epsilon}_{1:W} := (\boldsymbol{\epsilon}^{(1)}, \ldots, \boldsymbol{\epsilon}^{(W)})$. Note that the noise prior is applied during both training and sampling, enabling the model to learn how to deal with temporally correlated noise samples. A more comprehensive study of alternative noise priors in the context of ERDM is left to future work.

## D Experimental Details

### D.1 Dataset details

**Licenses.** We use the Navier-Stokes dataset with four obstacles introduced by Otness et al. [41]. Its license is CC BY 4.0. We also use the ERA5 dataset provided by Weatherbench-2 [46]. Its license is CC BY 4.0.

**ERA5 variables.** We use the $1.5°$ resolution of the data (a $240 \times 121$ grid) provided by [46], with 69 prognostic (input and output) variables: Temperature (t), geopotential (z), specific humidity (q), and the u and v components of wind (u,v) over 13 pressure levels as well as the four surface variables 2m temperature (2t), mean sea level pressure (mslp), and u and v components of winds at 10m (10u, 10v). Following Weatherbench-2, the 13 levels are $\{50, 100, 150, 200, 250, 300, 400, 500, 600, 700, 850, 925, 1000\}$. Numbers after any variable abbreviation refer to the level in hPa. For example, z500 refers to geopotential at 500 hPa. Besides these prognostic variables, we also use six static maps as additional inputs: A land-sea mask, soil type, orography, and $\sin(lat), \cos(lat)\cos(lon), \cos(lat)\sin(lon)$, where $lat, lon$ are the latitude and longitude of each location. We train our models on 12-hourly data, as in [45], from 1979 to 2020 and evaluate the models based on 64 initial conditions at 00/12 UTC, evenly spaced in 2021.

**Sizes.** The Navier-Stokes training set consists of around 6100 samples, and the corresponding validation set consists of 118 samples (slightly less for $W > 1$). We use the ERA5 training data

set at 12-hourly resolution with possible initial condition times in 00/06/12/18 UTC. The training data size is thus around 61000 samples. Following Weatherbench-2, we only use 00/12 UTC start times for evaluation. We use 64 samples for evaluation. During validation, we use ensemble sizes of $M = 10$ and $M = 5$ for Navier-Stokes and ERA5 respectively, which we increase to $M = 50$ and $M = 10$ during final testing.

## D.2 Implementation details and hyperparameters

**Architecture.** For all experiments and baselines that we train, we use the same basic ADM U-Net [13] architecture, only modified where necessary depending on the method. We always use 4 down- and up-sampling blocks, with channel multipliers $1, 2, 3, 4$. The base channel dimension is 64 for Navier-Stokes and 256 for ERA5 experiments. We use attention layers at the two lowest spatial resolutions. We use a dropout rate of 0.15 for Navier-Stokes, but disable it for ERA5. Other architectural hyperparameters are kept to the defaults from EDM. This results in a parameter count of 29.5 million for Navier-Stokes, and 517 million for ERA5. For the models operating on sequence, ERDM and EDM with $W > 1$, we integrate causal temporal attention layers into the 2D U-Net before each down- and up-sampling block. This increases the respective parameter counts slightly to 31.5 and 537 million for Navier-Stokes and ERA5, respectively. For ERA5, we use circularly padded convolutions along the longitude dimension (and zero-padding for the latitude dimension) to better respect the spherical structure of the Earth.

**Navier-Stokes hyperparameters.** We train each model for 300 epochs with a global effective batch size of 32, leading to almost $57,000$ gradient steps. The effective batch size is kept the same by balancing per-GPU batch size and gradient accumulation steps appropriately over different training runs. We use a cosine learning rate schedule with a linear warmup period of 10 epochs and a peak learning rate of $6 \times 10^{-4}$. We use the AdamW optimizer with a weight decay of $10^{-4}$. During evaluation, we use an exponential moving average (EMA)–based on an EMA decay rate of 0.995– version of the model. Because the input spatial grid, $221 \times 42$, after four halvings does not result in an even number–a problem for the residual connections in the upsampling blocks–we bilinearly interpolate it to a $256 \times 64$ grid at the input layer (and back before the final convolution layer).

**ERA5 hyperparameters.** We use a cosine learning rate schedule with a linear warmup period of 5 epochs and a peak learning rate of $5 \times 10^{-4}$. ERDM and EDM are trained with a global effective batch size of 32 and 512, respectively. The difference in this choice was informed by how many data points we could fit into GPU memory (ERDM has higher memory needs due to operating on a window of data) and the observation that EDM benefited more strongly from higher effective batch sizes. The maximum length of the schedule is 60 epochs for ERDM and 1000 epochs for EDM, resulting in a comparable number of total gradient steps. We use the AdamW optimizer without weight decay, clipping gradients above a norm of 0.8. During evaluation, we use an EMA version of the model, with an EMA decay rate of 0.9999. Because the input spatial grid, $240 \times 121$, after four halvings does not result in an even number–a problem for the residual connections in the upsampling blocks–we bilinearly interpolate it to a $240 \times 128$ grid at the input layer (and back before the final convolution layer).

## D.3 Baseline details

**DYffusion.** We follow the default values used by DYffusion for its Navier-Stokes experiments, but replace the interpolator and forecaster architecture with the more performant U-Net and use our optimization hyperparameters, both described above. We tuned the interpolator dropout rates and found a rate of 0.5 to produce the best results. The interpolator dropout rate is a key hyperparameter in DYffusion, since it is responsible for the stochasticity in the model (if 0, it reduces to a deterministic model). As for ERDM, we use a window size of $W = 6$. Our improved architecture and training hyperparameters jointly improved the DYffusion CRPS scores by over $3\times$ compared to the results in the original paper.

**PDE-Refiner** PDE-Refiner [32] is a next-step diffusion-derived forecaster. We tune the two key hyperparameters of PDE-Refiner, $K$ and $\sigma_{\min}$. The best forecasts were obtained with $K = 6, \sigma_{\min} = 0.002$. The recommended values from [32], $K = 4, \sigma_{\min} = 2e\text{-}7$, did not work as well. We use the same denoiser architecture as for EDM. PDE-Refiner performs extremely well for the first few

timesteps of the Navier-Stokes rollout, but rapidly diverges after that. We believe this is because it cannot model temporal interactions.

**EDM.** Our primary focus is to evaluate ERDM's performance relative to common EDM-derived approaches for dynamics forecasting. Consequently, our key baseline is an EDM denoiser parameterized by $D_\theta(x_1 \ldots, x_W; \sigma, y_0)$, where all $x_w$ are corrupted according to the same noise standard deviation, $\sigma$. With $W = 1$, we recover a next-step forecasting conditional EDM as in [45, 29]. On Navier-Stokes, we tried $W \in \{1, 2, 4, 6\}$, and found $W = 4$ to work best for EDM. For computational reasons, we only report an EDM run with $W = 1$ for ERA5. To condition the denoiser network on the initial condition, $y_0$, we concatenate it along the channel dimension to the input, noisy window, $x_{1:W}$. When $W > 1$, $y_0$ is thus first duplicated along the window dimension. We found this to perform better than operating on a $W + 1$ window, where $y_0$ is inserted into the first position of the window. Preconditioning is applied before concatenation, *only* to the noisy window.

To be clear, the EDM "video" baseline with $W > 1$ does not use rolling diffusion. It is a standard sequence-to-sequence EDM, using the same 3D denoiser architecture as ERDM, that operates autoregressively. Conditioned on a single clean frame (e.g., at time $t$), it jointly predicts a window of future frames ($t + 1$ to $t + W$). Unlike ERDM, all frames in this window are denoised from the same initial noise level. To generate a long forecast, the last frame of the predicted window (at time $t + W$) is then used as the new "clean" condition for the next prediction step.

### D.4 Distinctions between GenCast and our EDM model for ERA5

Our primary baseline, conditional EDM, applied to ERA5 shares several similarities with GenCast [45]. A primary distinction lies in the spatial resolution: GenCast is trained on a spatial resolution of $1°$ and $0.25°$, whereas our model, EDM, and other baselines are exclusively trained at a $1.5°$ resolution. The denoiser network architecture presents another significant divergence. GenCast uses a graph neural network, similar to the approach in Lam et al. [30], while we use a 2D U-Net for our EDM baseline borrowed from [26, 13]. Furthermore, GenCast incorporates SST as both an input and an output variable. sea surface temperature as an input and output, which we do not. More minor differences between GenCast vs. our EDM are: ODE solver (DPMSolver++ vs. Heun), training noise level distribution (uniform vs. lognormal), noise distribution (spherical vs. i.i.d.), optimization hyperparameters, and residual vs. full-state prediction.

### D.5 Compute resources

For Navier-Stokes experiments, we trained all models and baselines on $2 - 8$ L40S or A100 GPUs. For ERA5, we trained some models (including our final ERDM version) on 4 H200 GPUs, but performed most of our development work on $8 - 16$ A100 GPUs (per training run). For ERDM with $W = 6$ ($W = 4$), a full training run on 8 L40S GPUs took around 16 (10.5) hours. For ERA5 with $W = 6$, a full training run on 4 H200 GPUs took almost 5 days.

### D.6 Metrics

Let $y \in \mathbb{R}^{I \times J}$ indicate the targets for a specific time step, and $\hat{y} \in \mathbb{R}^{M \times I \times J}$ the corresponding predictions with an ensemble size of $M$. $I$ and $J$ are the size of the height dimension and width dimension, respectively. For ERA5, these are latitude and longitude, respectively. The definitions of the metrics below follow common practices (e.g., cf. Weatherbench-2 [46]).

**Weighting.** For ERA5 experiments, we follow the common practice of area-weighting all spatially aggregated metrics according to the size of the grid cell, ensuring that they are not biased towards the polar regions [46]. The unnormalized area weights are computed as $\tilde{w}(i) = \sin \phi_i^u - \sin \phi_i^l$, where $\phi_i^u$ and $\phi_i^l$ represent upper and lower latitude bounds, respectively, for the grid cell with latitude index $i \in \{1, 2, \ldots, I\}$. The normalized area weights, used in all metrics below are thus $w(i) = \frac{\tilde{w}(i)}{\frac{1}{I} \sum_{i=1}^{I} \tilde{w}(i)}$, where $I$ is the number of latitude indices. For the Navier-Stokes experiments, we report unweighted spatially aggregated metrics (i.e., $w(i) = 1$ for all $i$).

**Ensemble-mean RMSE.** In ensemble-based forecasting, the RMSE is typically computed based on the ensemble mean prediction as follows:

$$\mathrm{RMSE}_{\mathrm{ens}} = \sqrt{\frac{1}{IJ}\sum_{i,j}w(i)(\mathrm{mean}_m(\hat{\boldsymbol{y}}_{m,i,j}) - \boldsymbol{y}_{i,j})^2}, \tag{25}$$

where $\mathrm{mean}_m$ refers to the average over the ensemble dimension.

**Spread-skill ratio (SSR).** Following Fortin et al. [16], the spread-skill ratio is defined as the ratio between the ensemble spread and the ensemble-mean RMSE. The ensemble spread is defined as the square root of the ensemble variance. Thus, the SSR is defined as:

$$\mathrm{Spread} = \sqrt{\frac{1}{IJ}\sum_{i,j}w(i)\mathrm{var}_m(\hat{\boldsymbol{y}}_{m,i,j})}, \qquad \mathrm{SSR} = \sqrt{\frac{M+1}{M}}\frac{\mathrm{Spread}}{\mathrm{RMSE}_{\mathrm{ens}}}, \tag{26}$$

where $\mathrm{var}_m$ refers to the variance over the ensemble dimension, and $\sqrt{(M+1)/M}$ is a correction factor which is especially important to include for small ensemble sizes. The SSR serves as a simple measure of the reliability of the ensemble, where values closer to 1 are better. Values smaller (larger) than 1 indicate underdispersion (overdispersion). In our ablations, we also report the average squared deviation, $\overline{(1-\mathrm{SSR})^2}$ (lower is better), where the overline indicates averaging over all rollout/lead times. This is more useful than reporting $\overline{\mathrm{SSR}}$, since such a metric would be sensitive to canceling out deviations from 1 on both sides (e.g., an SSR of 0.5 and 1.5 on the first and last half of the rollout, respectively, would average to 1).

**Continuous ranked probability score (CRPS).** Following [59, 46], we use the unbiased version of the CRPS [37] (lower is better)

$$\mathrm{CRPS} = \frac{1}{IJ}\sum_{i,j}w(i)\left[\frac{1}{M}\sum_{m=1}^{M}|\hat{\boldsymbol{y}}_{m,i,j} - \boldsymbol{y}_{i,j}| - \frac{1}{2M(M-1)}\sum_{m=1}^{M}\sum_{n=1}^{M}|\hat{\boldsymbol{y}}_{m,i,j} - \hat{\boldsymbol{y}}_{n,i,j}|\right]. \tag{27}$$

**CRPS skill score (CRPSS).** The CRPSS is commonly used to compare the CRPS scores relative to a baseline. It is computed as $1 - \mathrm{CRPS}_{\mathrm{model}}/\mathrm{CRPS}_{\mathrm{baseline}}$, where $\mathrm{CRPS}_{\mathrm{model}}$ and $\mathrm{CRPS}_{\mathrm{baseline}}$ refer to the CRPS scores for the evaluated model and baseline, respectively. Values larger than $0$ indicate that the model is better than the baseline model in underscore, while values smaller than 0 indicate that the model is worse than the baseline model.

## E  Ablations

Starting from our proposed ERDM configuration ("Base" in Table 2), which uses the second-order Heun sampler, loss weighting with $P_{\mathrm{mean}} = 0.5, P_{\mathrm{std}} = 1.2$, a temporally correlated progressive noise prior following [18] with $\epsilon_{\mathrm{prog}} := \alpha = 1$ (see Appendix C.4), noise schedule parameters $\sigma_{\mathrm{min}} = 0.002, \sigma_{\mathrm{max}} = 200, \rho = -10$, and a window size of $W = 6$, we systematically varied individual components. For sampling, we use $N = 1.25$ ($\Delta t = 0.8$) and $S_{\mathrm{churn}} = 0$ by default. The first window is initialized based on predictions from the EDM $W = 4$ baseline. Our findings, summarized in Table 2, highlight several critical elements for achieving optimal performance.

**Heun versus Euler sampler.** Compared to the first-order Euler method (Algorithm 2), the second-order sampling algorithm detailed in Algorithm 3 achieved superior performance in CRPS, MSE, and SSR metrics. This improvement comes at the cost of doubling the neural function evaluations due to the correction step, which significantly reduces sampling speed. Nevertheless, we deemed this increased computational overhead justifiable for the enhanced accuracy.

**Number of sampling steps.** Our base ERDM model uses $N = 1.25$ sampling steps "per snapshot", which results in a step size of $\Delta t = 1/N = 0.8$. That is, depending on the noise levels at the start of an iteration for solving the ODE, it may use 1 or 2 sampling steps before outputting the first frame and sliding the window. Increasing $N$ extends sampling time, but optimal performance is found at a "sweet spot"; for example, $N = 1.25$ outperforms $N = 2$ for Navier-Stokes. This was also confirmed for ERA5, where CRPS scores showed minimal improvement for $N > 2$ (with an optimum around $N = 2$). The parameter $N$ is also crucial for tuning the model's spread—smaller $N$ increases spread, larger $N$ reduces it—thereby providing a control for adjusting the spread-skill ratio. Ultimately, the ideal $N$ and step size depend on the dataset complexity and the window size, where larger window sizes generally tolerate larger step sizes (smaller $N$).

Table 2: Ablation study results on the Navier-Stokes dataset. The most important design choices, without which ERDM would degrade to worse performance than the EDM baseline (highlighted in red), are 1) a proper 3D architecture; 2) an appropriate noise schedule (in particular, not EDM's default $\rho = 7$); 3) our proposed loss weighting; 4) a fixed training schedule. CRPSS refers to the CRPS skill score, where $> 0$ means that the model is better than the baseline model in underscore. CRPSS values smaller than 0 indicate that the model is worse than the baseline model. For all other metrics, lower is better.

| Ablation | $\overline{\mathrm{CRPS}}_{\times 10^2}$ | $\overline{\mathrm{MSE}}_{\times 10^3}$ | $\overline{\mathrm{CRPSS}_{\mathrm{ERDM}}}$ | $\overline{\mathrm{CRPSS}_{\mathrm{EDM}}}$ | $\overline{(1-\mathrm{SSR})^2}$ |
|---|---|---|---|---|---|
| Base | 0.904 | **0.588** | 0.000 | 0.432 | 0.052 |
| **Sampling** | | | | | |
| Euler Sampler | 1.160 | 0.752 | -0.279 | 0.288 | 0.227 |
| $\Delta t = 1$ | 1.270 | 0.810 | -0.399 | 0.224 | 0.343 |
| $\Delta t = 0.9$ | 1.067 | 0.823 | -0.160 | 0.349 | 0.056 |
| $\Delta t = 0.7$ | 0.986 | 0.668 | -0.084 | 0.389 | 0.080 |
| $\Delta t = 0.5$ | 1.042 | 0.648 | -0.159 | 0.349 | 0.121 |
| **Loss Weighting** | | | | | |
| No $f(\sigma)$ | 2.168 | 2.987 | -1.296 | -0.243 | 0.244 |
| $P_{\mathrm{mean}} = 1$ | 1.082 | 0.783 | -0.181 | 0.338 | 0.057 |
| $P_{\mathrm{mean}} = 0$ | 1.431 | 1.517 | -0.489 | 0.177 | 0.119 |
| $P_{\mathrm{mean}} = -1.2$ | 2.001 | 2.694 | -1.086 | -0.134 | 0.259 |
| **Noise Prior ($\epsilon$)** | | | | | |
| $\epsilon_{\mathrm{indep}}$ | 1.176 | 0.977 | -0.240 | 0.309 | 0.134 |
| $\epsilon_{\mathrm{prog}=2}$ | 1.060 | 0.715 | -0.151 | 0.357 | 0.065 |
| $\epsilon_{\mathrm{prog}=10}$ | 1.404 | 1.353 | -0.474 | 0.190 | 0.098 |
| **Noise schedule** | | | | | |
| Random training sched. | 1.904 | 1.998 | -1.065 | -0.130 | 0.347 |
| $\sigma_{\mathrm{min}} = 0.001$ | 1.040 | 0.758 | -0.102 | 0.383 | 0.062 |
| $\sigma_{\mathrm{min}} = 0.01$ | 1.511 | 1.342 | -0.692 | 0.052 | 0.219 |
| $\sigma_{\mathrm{max}} = 80$ | 1.094 | 0.878 | -0.175 | 0.346 | 0.080 |
| $\sigma_{\mathrm{max}} = 800, \rho = -20$ | 1.249 | 1.086 | -0.355 | 0.244 | 0.056 |
| $\rho = -5$ | 1.529 | 1.569 | -0.615 | 0.121 | 0.189 |
| $\rho = -20$ | 1.060 | 0.695 | -0.180 | 0.339 | **0.045** |
| $\rho = 7$ | 1.982 | 2.382 | -1.273 | -0.265 | 0.098 |
| **Window Size ($W$)** | | | | | |
| $W = 4$ | 1.411 | 1.308 | -0.487 | 0.185 | 0.104 |
| $W = 8$ | 1.258 | 1.134 | -0.313 | 0.276 | 0.196 |
| $W = 12$ | 1.709 | 1.841 | -0.765 | 0.041 | 0.422 |
| **Architecture** | | | | | |
| Uncond. Pre-training | 1.475 | 1.732 | -0.528 | 0.157 | 0.074 |
| 2D arch. | 3.931 | 2.979 | -3.202 | -1.209 | 3.476 |
| **Initialization** | | | | | |
| Init=EDM $W = 1$ | 1.176 | 0.977 | -0.240 | 0.309 | 0.134 |
| Init=persistence | 14.789 | 172.233 | -25.361 | -20.412 | 0.765 |
| Init=truth | **0.899** | 0.593 | **0.053** | **0.492** | 0.286 |

**Loss weighting.** As detailed in the main text, a well-tuned loss weighting can significantly improve results. In particular, using no loss reweighting based on the PDF of the lognormal distribution, $f(\sigma; P_{\text{mean}}, P_{\text{std}})$ ("No $f(\sigma)$"), degraded CRPS and MSE scores by more than $2\times$ and $5\times$, respectively. It also resulted in a worse calibration score. Similarly, choosing a poor value for the center of the lognormal distribution, e.g., $P_{\text{mean}} = -1.2$, resulted in almost as suboptimal results. Interestingly, $P_{\text{mean}} = -1.2$ corresponds to the default value in EDM, but we found values $\geqslant 0.5$ to be necessary for ERDM. We found EDM's default $P_{\text{std}} = 1.2$ to work well for ERDM generally.

**Noise prior.** As hinted in Appendix C.4, we found it beneficial to use a temporally correlated noise prior as opposed to simply sampling i.i.d. noise for each snapshot ("$\epsilon_{\text{indep}}$"). The noise prior that we use [18] introduces one hyperparameter, $\alpha$, which balances the strength of the correlation. We generally found $\alpha \in [0.5, 1.5]$ to work best. Similarly to the findings of Ge et al. [18], using overly large $\alpha$ (e.g., $\alpha = 10$, which introduces large correlations between noise snapshots) is detrimental to performance and worse than simply using i.i.d. noise.

**Other design choices.** Surprisingly, unconditional static pre-training of the non-temporal components of the model before fine-tuning on the forecasting task was found to degrade performance ("Uncond. Pre-training"). Preliminary ERA5 experiments seemed to confirm this finding. Moreover, we did not find ERDM to be overly sensitive to the window size within a reasonable range ($W = 6$ as base). However, overly large window sizes (e.g., $W = 12$) significantly degrade performance, potentially due to fixed network capacity. Note that for each window size, we tuned the sampling step size from $\Delta t \in \{0.25, 0.5, 0.8, 1\}$ and report the best results only.

## E.1 ERDM initialization

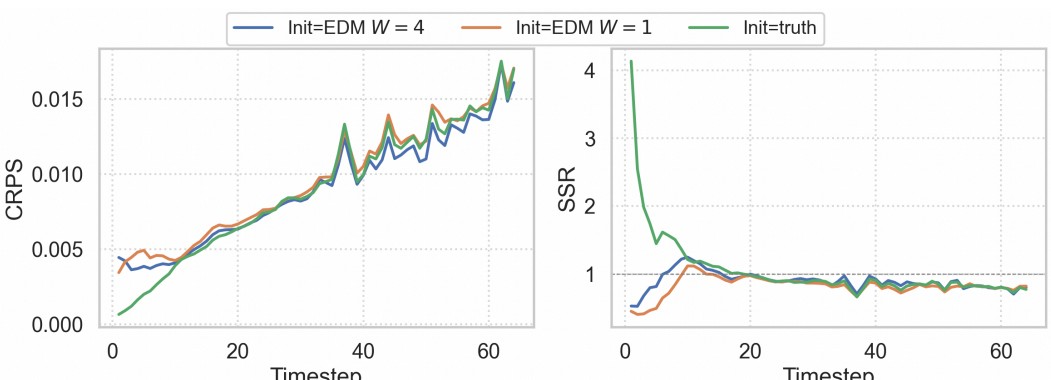

Figure 7: Impact of first-window initialization methods on ERDM's Navier-Stokes forecast performance. The chosen initialization strategy significantly affects short-range skill (up to timestep 10), but its influence diminishes over longer forecast horizons. While initializing with ground truth data (an impractical approach) unsurprisingly yields the best initial results, it also produces an artificially inflated Spread-Skill Ratio (SSR) due to a near-zero initial RMSE. Notably, this early advantage dissipates throughout the rollout, leading to final CRPS and SSR scores that are largely indistinguishable regardless of the initialization method.

In Fig. 7 and the last part of Table 2, we ablate possible initialization schemes for ERDM. These include two neural initializations based on the EDM baselines with $W = 1$ and $W = 4$ (our default for Navier-Stokes being EDM $W = 4$), alongside two simple non-neural approaches. One non-neural method, Init=truth, initializes ERDM using ground truth data for the first window. While clearly impractical, it serves as an informative upper bound on the achievable performance of ERDM. The other, Init=persistence, populates the first window by duplicating the initial condition $y_0$; this approach performed very poorly and is omitted from Fig. 7 for visual clarity. Expectedly, Init=truth yields optimal initial CRPS results but cannot be used in practice. More significantly, Fig. 7 demonstrates that the choice of initialization predominantly impacts ERDM's short-range forecast skill (up to approximately timestep 10). The initial performance advantages conferred by methods like Init=truth, or differences among other practical schemes, diminish substantially as the forecast horizon extends. By the end of the rollouts, both CRPS and SSR scores become largely indistinguishable across the various practical initialization methods, including our default EDM $W = 4$

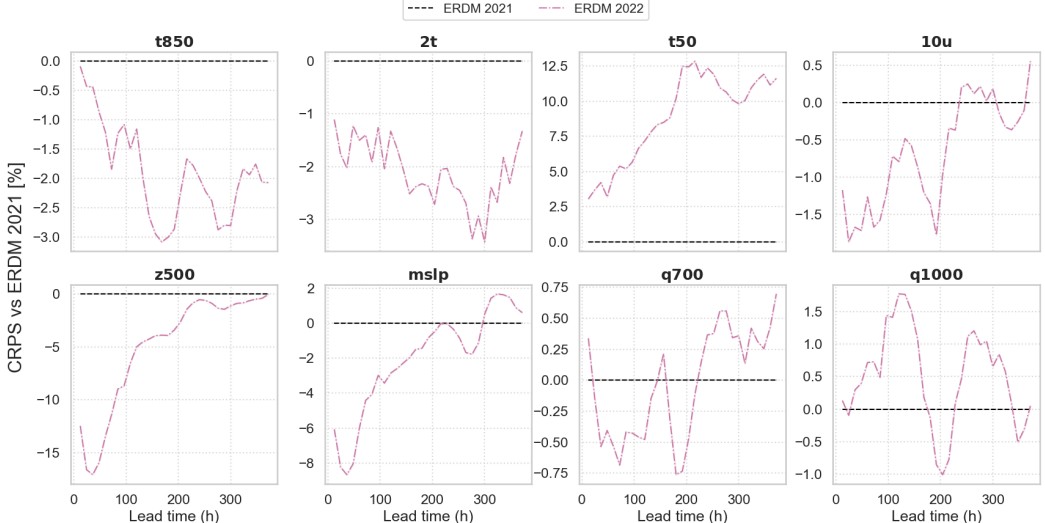

Figure 8: Impact of ERA5 evaluation year on ERDM's CRPS performance. The model, trained on data up to 2020 (inclusive), generally shows improved CRPS scores when evaluated on initial conditions from 2022 compared to 2021 (the baseline year used throughout this paper), with `t50` being an exception. While most variables exhibit CRPS changes of less than 3%, geopotential and mean sea level pressure are significantly more sensitive to the evaluation year, particularly for lead times under 8 days (e.g., `z500` scores improve by up to 15%). This variability underscores the need for careful interpretation when comparing model results from different evaluation years, especially for geopotential and stratospheric fields.

initialization. This convergence suggests that while the first-window initialization is crucial for early forecast accuracy, its specific choice becomes less critical for the long-range performance of ERDM (except when using a naive initialization such as `persistence`).

### E.2 ERA5 evaluation year ablation

In our quantitative comparison, we evaluated ERDM, EDM, and IFS ENS on the same 64 initial conditions evenly spaced in 2021, ensuring a fair comparison. Unfortunately, we only have access to the NeuralGCM ENS and Graph-EFM results for 2020. This discrepancy highlights the importance of understanding how model scores might vary across different evaluation years. We investigate this for our ERDM model (trained on data up to 2020 inclusive) in Fig. 8. As detailed in the figure and its caption, ERDM's CRPS performance indeed shows sensitivity to the evaluation year. For example, when evaluated on 2022 initial conditions, scores often improved relative to our 2021 baseline, though for most variables this change was less than 3%. However, certain variables, notably geopotential (with `z500` improving by up to 15%) and mean sea level pressure, exhibited significantly greater sensitivity, particularly at lead times under 8 days. These findings suggest that while our 2021-based evaluations provide a consistent benchmark for ERDM, EDM, and IFS ENS, direct comparisons with results from NeuralGCM ENS and Graph-EFM on 2020 data should be made with an awareness of this potential inter-annual performance variability.

## F    Additional Results

### F.1    Additional Navier-Stokes results

**MSE scores.** For completeness, we report the ensemble-mean MSE test rollout scores for all Navier-Stokes models in Fig. 9, analogously to our main CRPS and SSR results in Fig. 4. The MSE scores are strongly correlated to the CRPS scores, which is why we deferred their discussion to the appendix. Similarly to the CRPS scores, ERDM achieves significantly better MSE scores, especially for long-range lead times.

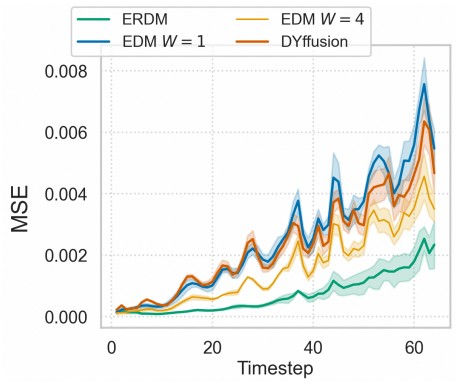

Figure 9: Ensemble-mean MSE scores for the Navier-Stokes test rollout.

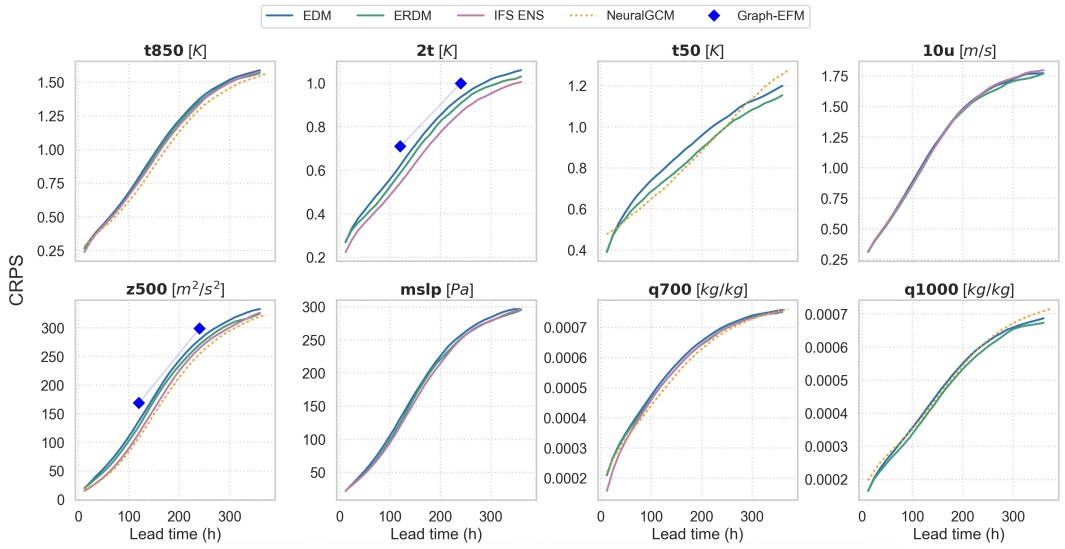

Figure 10: Absolute CRPS corresponding to the relative scores shown in Fig. 5. Lower is better.

A slight difference to the CRPS results is that the relative improvement of the block-wise autoregressive EDM $W = 4$ baseline over the DYffusion baseline is larger in terms of MSE than CRPS.

### F.2    Detailed ERA5 quantitative results

A comprehensive quantitative assessment of forecast skill is presented in Fig. 11, which displays the CRPS for ERDM, IFS ENS, NeuralGCM, and Graph-EFM, relative to the EDM baseline. These relative CRPS values are reported as percentages (where lower indicates better performance) for numerous atmospheric variables (extending the set shown in Fig. 5) over the full 15-day forecast horizon. These results underscore that ERDM consistently achieves significantly lower CRPS values than the EDM baseline. The corresponding absolute CRPS scores are visualized in Fig. 10, and a selection of them are summarized in Table 3. A full relative comparison between ERDM and EDM is presented in the form of a scorecard in Fig. 13. A quantitative assessment of forecast calibration is presented in Fig. 12, which displays the spread-skill ratio for the same set of models and variables. ERDM generally generates well-calibrated forecasts on par or better than IFS ENS, and consistently more calibrated forecasts than the EDM baseline or NeuralGCM.

Table 3: Tabular CRPS scores for a selection of forecast horizons (three, seven, and ten days) and variables (10m u-component of wind and 500 hPa geopotential). Lower is better.

| Model | u10m | | | z500 | | |
|---|---|---|---|---|---|---|
| | 3d | 7d | 14d | 3d | 7d | 14d |
| EDM | 0.6923 | 1.324 | 1.769 | 76.04 | 204.9 | 327.4 |
| ERDM | **0.6748** | **1.312** | **1.737** | 69.29 | 197.5 | 316.7 |
| IFS ENS | 0.6830 | 1.318 | 1.785 | 58.41 | 183.4 | 317.3 |
| NeuralGCM | | N/A | | **54.59** | **173.1** | **311.1** |

### F.3    Comprehensive ERA5 power spectra

To assess the physical realism of the generated forecasts, we analyze their spectral properties, as presented in Fig. 14. This figure shows the spectral density of 14-day forecasts for several key atmospheric variables, averaged over high latitudes ($[60°, 90°]$). The absolute spectra (subfigure (a)) and spectra normalized by ERA5 reanalysis (subfigure (b)) consistently demonstrate the high fidelity of ERDM. Our model's spectra closely align with the ERA5 target across a broad range of zonal wavenumbers and frequently match or even slightly outperform those from the operational

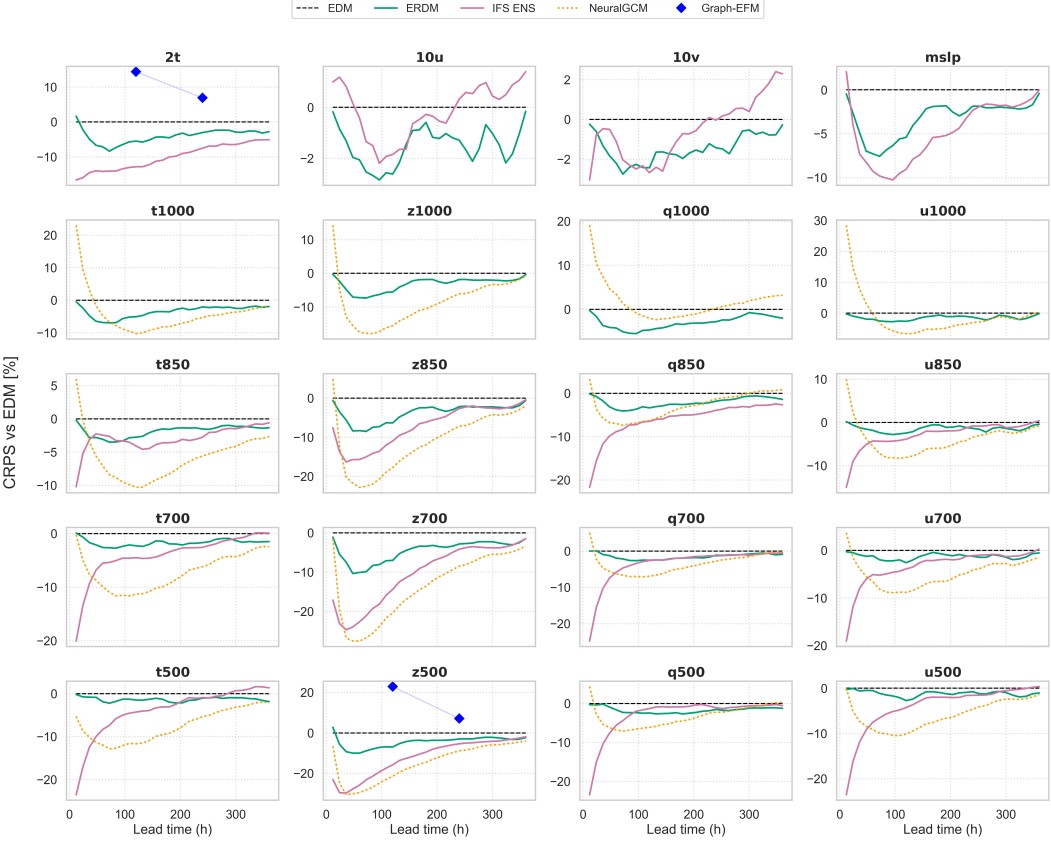

Figure 11: CRPS relative to next-step autoregressive EDM baseline (in %; lower is better) for 15-day rollouts across a more comprehensive set of variables than in Fig. 5.

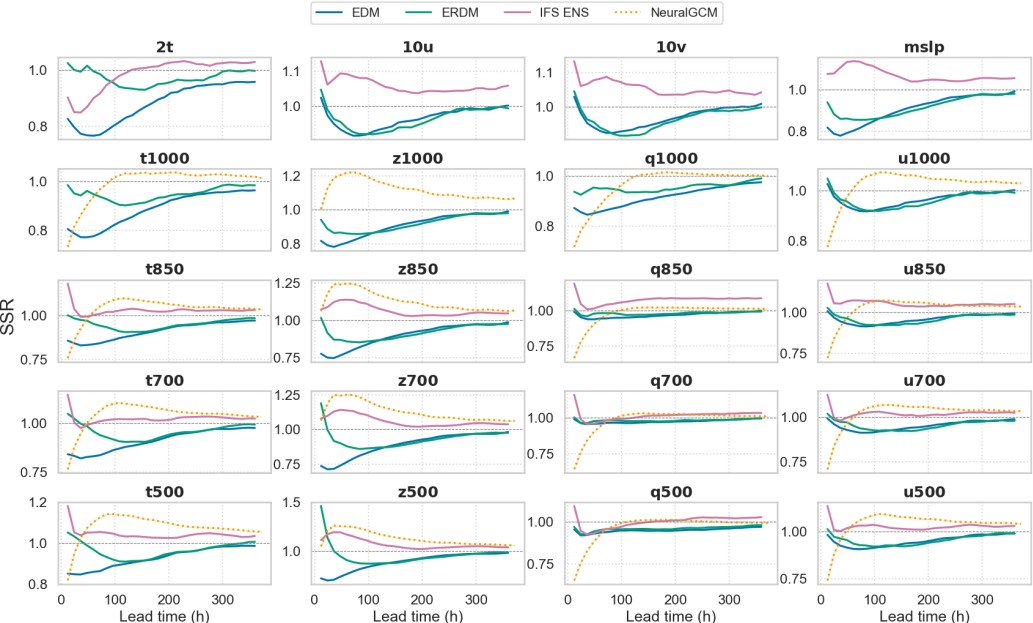

Figure 12: Spread Skill Ratio (SSR) computed as the ensemble spread / ensemble-mean RMSE based on 10-member ensembles for each model. Closer to 1 is better and represents a well-calibrated ensemble. Among the ML-based methods, ERDM tends to produce the best calibration in terms of SSR, together with IFS ENS, except for the first few lead times of z500 and z700 where it is overcalibrated. NeuralGCM and EDM tend to be severely underdispersed, while IFS ENS tends to be slightly overdispersed, in the short range.

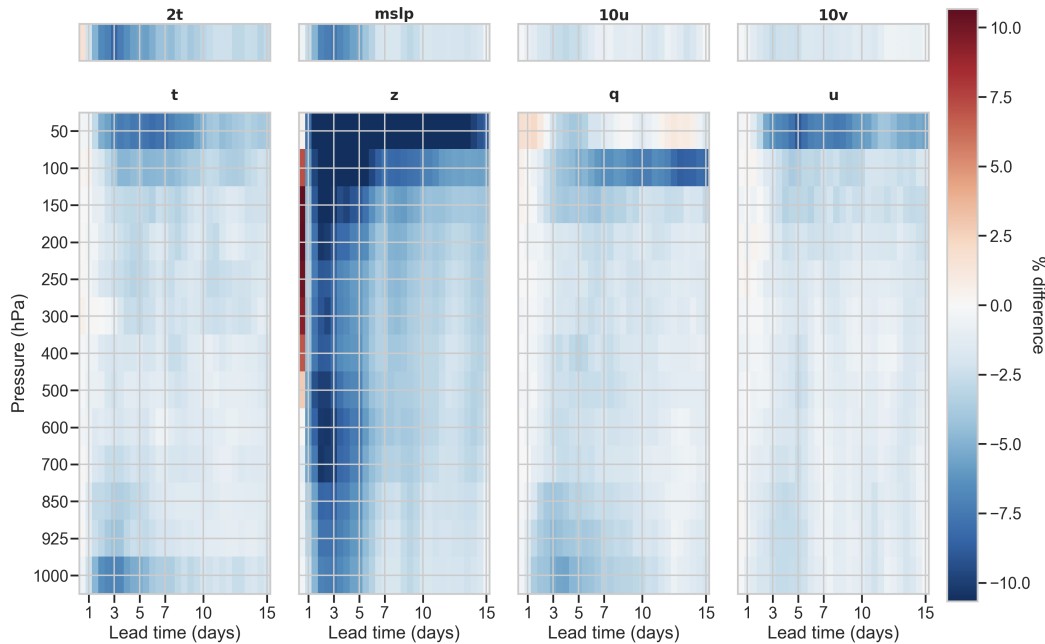

Figure 13: Scorecard of ERDM's CRPS relative to next-step autoregressive EDM baseline (in %; lower is better) for all predicted variables (except v, which is correlated with u), pressure levels, and lead times (up to 15 days). ERDM consistently outperforms the EDM baseline, especially for geopotential and high-altitude levels.

physics-based IFS ENS model. This accurate representation indicates that ERDM effectively captures the energy distribution from large planetary scales down to smaller synoptic scales for diverse atmospheric fields. In contrast, NeuralGCM ENS exhibits a marked underestimation of energy at mid to high frequencies for several variables, evident by its spectra decaying more rapidly in the absolute plots and falling below unity in the normalized plots at these scales. The robust spectral characteristics of ERDM across multiple variables underscore its capability to produce physically consistent and realistic long-range weather forecasts.

### F.4 ERA5 forecast visualizations

See Fig. 15 and Fig. 16 for visualizations of example forecasts by ERDM for a specific initial condition (2022-01-01) and several lead times. As suggested by the good power spectra (see section above), ERDM produces realistic forecasts, including at long lead times (e.g., 15 days).

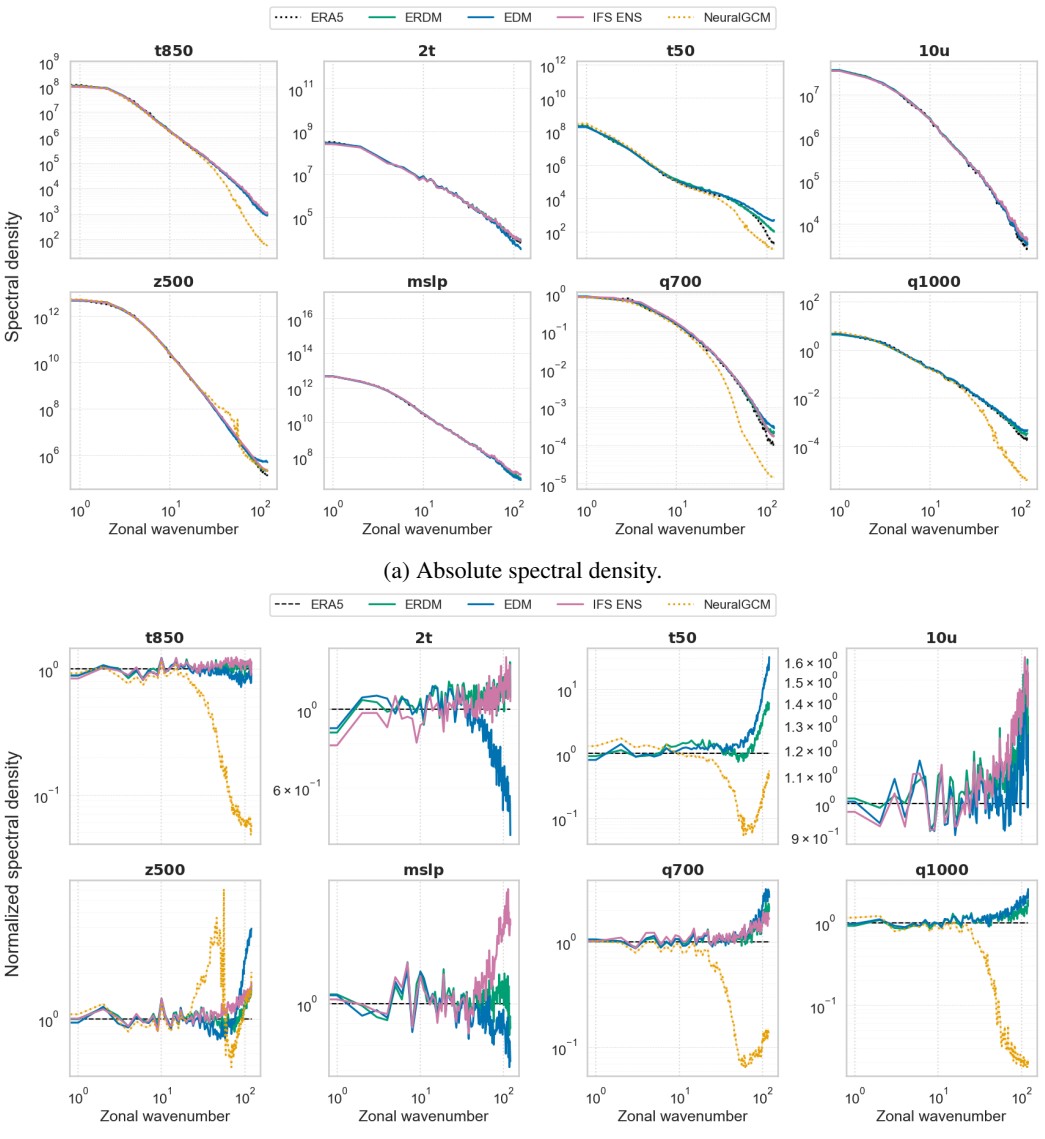

(a) Absolute spectral density.

(b) Normalized spectral density (relative to ERA5 reanalysis).

Figure 14: Spectral density of 14-day forecasts, averaged over high latitudes ($[60°, 90°]$). ERDM generates highly accurate spectra that match or even slightly beat the physics-based model IFS ENS, while NeuralGCM ENS underestimates energy at mid to high frequencies. Subfigure (a) displays the absolute spectral density, and subfigure (b) presents the normalized spectral density (absolute spectra divided by the target ERA5 spectra).

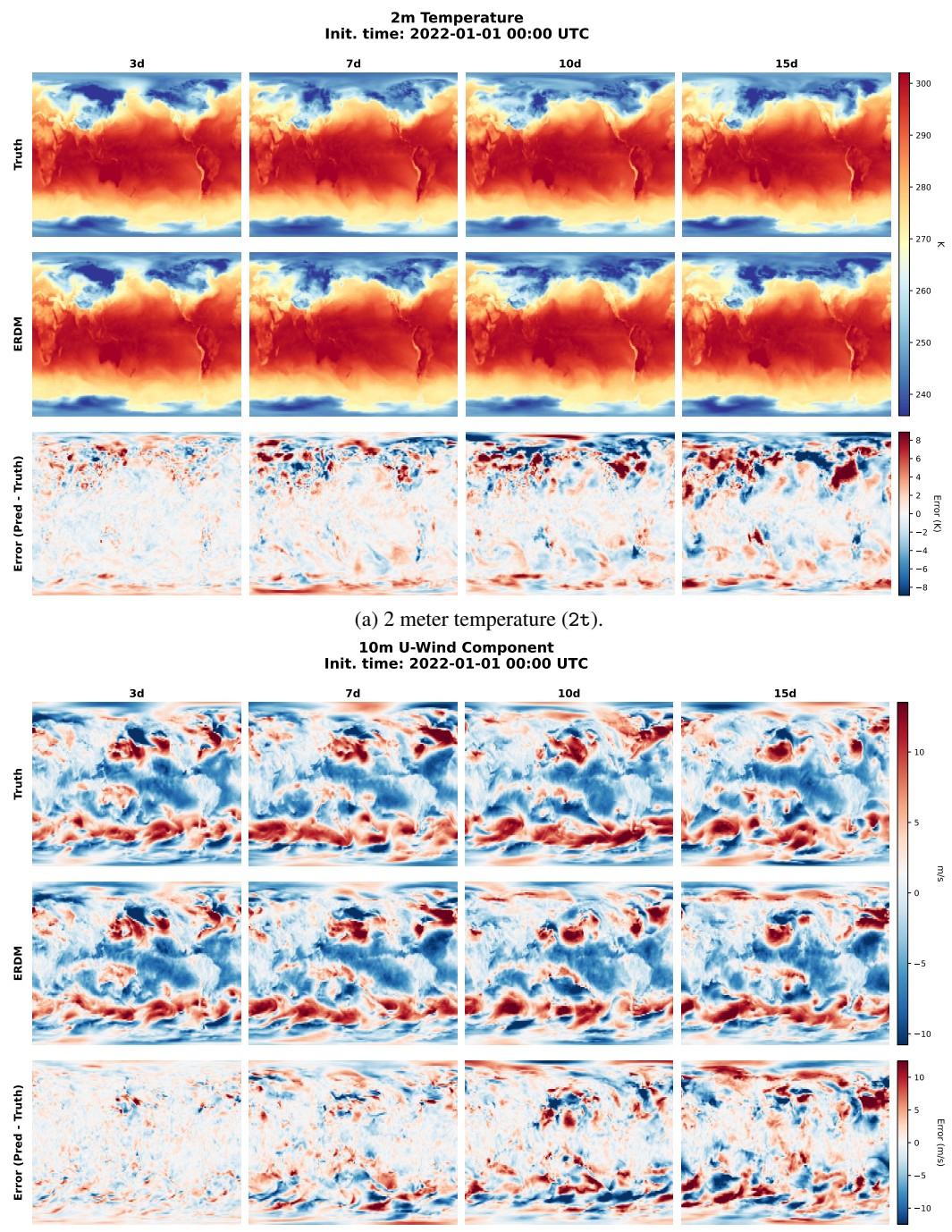

(a) 2 meter temperature (`2t`).

(b) 10m u component of wind (`10u`).

Figure 15: Example visualizations of ERDM (second row), the corresponding ground truth (first row), and the bias (last row) for two example variables and forecast lead times 3, 7, 10, and 15 days.

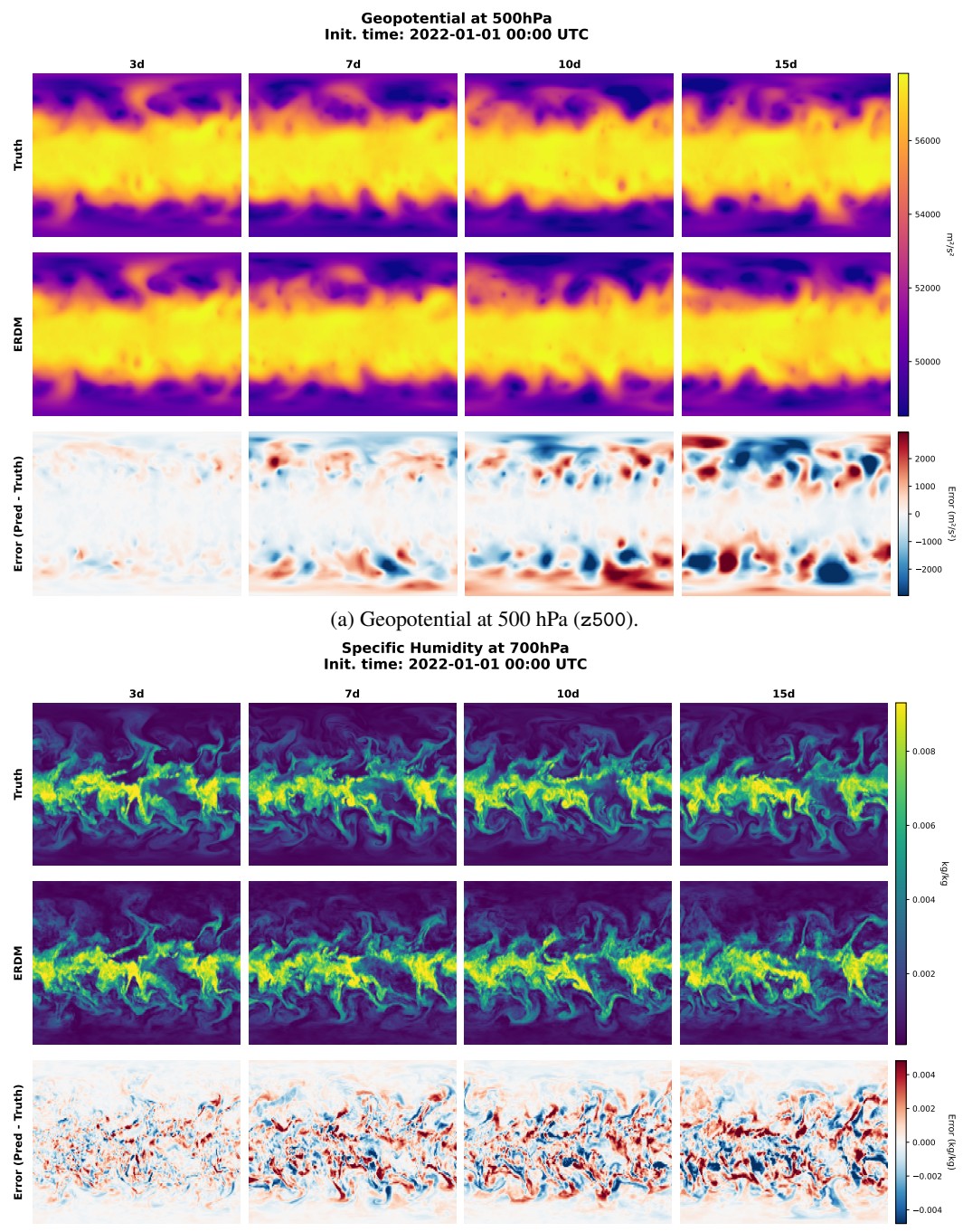

(a) Geopotential at 500 hPa (z500).

(b) Specific humidity at 700 hPa (q700).

Figure 16: Example visualizations of ERDM (second row), the corresponding ground truth (first row), and the bias (last row) for two example variables and forecast lead times 3, 7, 10, and 15 days.

