# OpenReview forum: "Elucidated Rolling Diffusion Models for Probabilistic Forecasting of Complex Dynamics"
_NeurIPS.cc/2025/Conference — NeurIPS 2025 poster_

### Official Review · Reviewer_jthi · 2025-06-24

**Clarity:** 3
**Significance:** 3
**Originality:** 2
**Rating:** 5
**Confidence:** 3

**Summary:**

The paper proposes Elucidated Rolling Diffusion Models (ERDM), which combines Rolling Sequence Diffusion with the Elucidated Diffusion Model (EDM) framework. The paper investigates various design choices, including hyperparameters and (denoising) model architecture. For the former, some modifications of the suggestions in the EDM paper are proposed. For the latter, temporal attention is added to the standard UNet model to acknowledge the spatio-temporal properties of the forecasting model. The method is tested on both Navier-Stokes fluid dynamics and weather forecasting.

**Questions:**

1.	The proposed method has a higher computational cost (per NFE) than a standard EDM since it operates on windows of lengths W. This is briefly acknowledged in the conclusion section, where it is mentioned that ERDM and EDM still have roughly the same computational cost in practice. However, this is not obvious from the empirical evaluation and I wonder if it would be possible to use the extra compute for EDM in a better way to improve it’s performance? In other words, does the observed improvement for ERDM primarily come from the fact that you effectively use W times the number of diffusion steps in ERDM (since each forecast is progressively denoised W times)?
2.	How are the forecasts produced with EDM, W=4? Does it also use rolling diffusion, or do you simply extract the first forecast in the window and throw away the rest?
3.	I found it a bit strange that you evaluate your method against NeuralGCM using different test sets. Even if you don’t have results for the different baselines for the same year (and don’t want to run them yourselves), could you not at least run the proposed method for both 2020 and 2021 and carry out two separate evaluations depending on the test sets for the different baselines?
4.	What’s the point of having two weighting functions $\lambda(\sigma)$ and $f(\sigma)$? You write (line 337) that it’s important to tune the $f(\sigma)$ function anyway, so I wonder how important the particular choice $\lambda(\sigma)*f(\sigma)$ is. Could you simply replace this product with some function $g(\sigma)$ that you tune based on empirical performance?
5.	It’s written (line 36) that you _“model the progressively increasing uncertainty [… by …] escalate noise for distant future snapshots”_. Is this really a correct interpretation? It is true that the model forecasts using higher noise levels at future time points (end of the W-window), but these time points are not completely denoised, so the model does not actually learn to forecast multiple steps into the future – correct? A better interpretation, in my opinion, would be to say that you produce 1-step forecasts by progressive denoising in multiple steps, in a way which is made possible by the temporal structure of the problem. When I read the intro (sentence above) and abstract I got the feeling that your proposed model could accurately simulate from the multi-step forecasting distribution $p(y_{t+W} | y_{t})$ in a way which respects the increased uncertainty with increasing W (in a principled way), but I don’t believe that this is the case..?
6.	Why do you have “weather forecasting” in the title when the method is more general and you evaluate it on one fluid dynamics problem and one weather forecasting problem?

**Ethical Concerns:**

["NO or VERY MINOR ethics concerns only"]

**Final Justification:**

The authors addressed W1, W2, Q1 in my original review to a satisfactory degree.

**Limitations:**

yes

**Quality:**

3

**Strengths And Weaknesses:**

The proposed method shows appears to have strong empirical performance compared to the baselines. In particular, it is shown that the ERDM outperforms a standard autoregressive EDM model, suggesting that there is a value in the rolling diffusion formulation of the probabilistic forecasting problem.

A weakness is that the contribution is somewhat limited, considering that is mainly consists of combining existing ideas for rolling diffusion with the EDM framework.

I’m also a bit sceptical about the word “Elucidating” in the title. I understand that this is inherited from the EDM paper, but I’m not convinced that this paper is “elucidating” how one should design diffusion model for forecasting in the best way. To me, the paper reads more like a typical methodological paper where the authors propose a particular method, which is evaluated empirically against alternative methods. Most design choices are taken from the EDM framework without shedding much additional light on these choices. This is perfectly fine, but I feel that the word “Elucidating” in the title promises more than what the paper delivers.

Regarding the resulting method, the proposed temporal UNet model used for denoising seems to be a key component, and perhaps this deserves to be emphasized more, at least if the focus is shifted from “elucidating the design space” to “proposing a method”.

---

> ### Author Rebuttal · Authors · 2025-07-31
>
> Thanks for your detailed review and insightful feedback. We're encouraged that you found our method to have strong empirical performance compared to the baselines.
>
> * **W1:** "the contribution is somewhat limited, considering that is mainly consists of combining existing ideas for rolling diffusion with the EDM framework."
>
> While we build upon established components, we respectfully argue that their successful integration into our ERDM framework is a non-trivial contribution tailored specifically for chaotic dynamical systems. The novelty lies in elucidating the specific adaptations required—which are not features of a standard EDM or RSDM—including a snapshot-dependent noise schedule, a bespoke temporal architecture, a specialized initialization strategy, and a tailored loss weighting.
>
> As our ablation studies demonstrate (Appendix, Table 1), these carefully engineered components are crucial for the final performance, showing that a naive combination of existing ideas would not have been sufficient. This is complemented by strong empirical results.
>
> * **W2:** "...sceptical about the word “Elucidating” in the title... I’m not convinced that this paper is “elucidating” how one should design diffusion model for forecasting in the best way... [it] promises more than what the paper delivers."
>
> We thank the reviewer for this fair point regarding the paper's title and its perceived promise. The primary reason for the name is make the link to the *Elucidated Diffusion Models (EDM)* framework, from which our method is derived, explicit; an alternative such as "EDM Rolling Diffusion Models" would be redundant.
>
> We agree that our paper is primarily methodological. Our use of "Elucidating" isn't meant to cover the entire design space, but to clarify the core principles—in architecture, initialization, and noise scheduling—needed to successfully adapt the EDM framework for dynamics forecasting.
>
> * **W3:** "...the proposed temporal UNet model used for denoising seems to be a key component, and perhaps this deserves to be emphasized more..."
>
> You're right, the temporal U-Net is essential to our method's success. We chose not to over-emphasize this specific architecture because our *primary contribution is the more general ERDM framework, which is agnostic to the specific backbone*.
>
> Our temporal U-Net could be swapped with any other capable 3D or sequential architecture, which might even yield better performance (see also discussion with reviewer yaoq). The key architectural takeaway isn't our specific U-Net, but that a capable temporal backbone is necessary, as a naive, non-temporal model would fail. We don't want to tie the ERDM framework too closely to one architectural choice, but rather highlight the principles that make it work.
>
> * **Q1:** "The proposed method has a higher computational cost (per NFE) than a standard EDM (...) does the observed improvement for ERDM primarily come from the fact that you effectively use W times the number of diffusion steps in ERDM...?"
>
> ***This is a misunderstanding***. ERDM actually needs *less* NFEs than EDM. However, each NFE is more costly for ERDM (due to the 3D vs. 2D denoiser architecture).
>
> The key difference is:
> * The standard autoregressive **EDM** must perform a full, expensive denoising schedule (with $N_{EDM} \approx 20$ steps) for *every single time step* it predicts.
> * In contrast, **ERDM's** denoises in parallel. The first snapshots of the window start in a state that is already "almost denoised" and only needs a very small number of refinement steps (`$N=2$` for our ERA5 experiments) to produce the next forecast before rolling forward.
>
> To make this relationship perfectly clear and quantitative, we will add the following *efficiency benchmark* to our revised paper.
>
> | Model | Inference NFEs | Inference Time (s) | Inference GPU Memory (GB) | Training GPU Memory (GB) |
> |:---|:---:|:---:|:---:|:---:|
> | EDM | 600 | 237 | 21 | 19 |
> | ERDM | 120 | 209 | 49 | 53 |
>
> *Legend: Efficiency metrics for ERDM vs. EDM. All measurements were performed on a single A100 GPU using mixed precision. Inference figures correspond to a 15-day (30 time steps), 5-member ensemble forecast with second-order Heun solver. Training uses a batch size of 1.*
>
> * **Q2:** "How are the forecasts produced with EDM, W=4? Does it also use rolling diffusion, or do you simply extract the first forecast in the window and throw away the rest?"
>
> This baseline does not use rolling diffusion. It is a standard *sequence-to-sequence EDM* (using the same 3D denoiser arch. as ERDM) that operates autoregressively. Conditioned on a single frame (e.g., at time $t$), it jointly predicts a window of future frames ($t+1$ to $t+W$). Unlike ERDM, all frames in this window are denoised from the same initial noise level. To generate a long forecast, the last frame of the predicted window (at time $t+W$) is then used as the new condition for the next prediction step. We will clarify this distinction in the revised manuscript.
>
> * **Q3:** "...could you not at least run the proposed method for both 2020 and 2021 and carry out two separate evaluations depending on the test sets for the different baselines?"
>
> We couldn't evaluate our model on the 2020 test set as it would be an unfair comparison; our model's training data includes the year 2020. Re-training our model with a different data split just for this comparison wasn't feasible due to the high computational cost of a full training run.
>
> * **Q4:** "What’s the point of having two weighting functions `$\lambda(\sigma)$` and `$f(\sigma)$`? ... Could you simply replace this product with some function `$g(\sigma)$` that you tune based on empirical performance?"
>
> Thank you for this insightful question. Our formulation using the product of $\lambda(\sigma)$ and $f(\sigma)$ was deliberately chosen to maintain consistency with the EDM framework, from which our method is derived. $\lambda(\sigma)$ is adopted directly from EDM, while our novel weighting $f(\sigma)$ is an adaptation of the EDM training noise distribution.
>
> We agree with your assessment that these could be merged into a single tunable function, $g(\sigma)$, and we believe exploring such alternative weighting schemes is a valuable direction for future work. We will add this clarification to the manuscript.
>
>
> * **Q5:** "...Is this really a correct interpretation? ...I got the feeling that your proposed model could accurately simulate from the multi-step forecasting distribution `$p(y_{t+W}|y_t)$` ... but I don’t believe that this is the case..?"
>
> Thank you for this excellent question. We agree that a complete denoising of the window to $t+W$ is not performed at each step. Our statement about "modeling progressive uncertainty" pertains to the effect of the noise schedule on the *joint denoising process* itself.
>
> By escalating noise for distant future snapshots, the framework compels the denoiser to treat information from these frames as less reliable. This process implicitly teaches the model that uncertainty grows over the forecast horizon. Therefore, the principle of increasing uncertainty is embedded within the model's learning mechanism. We will clarify this distinction in the revised manuscript.
>
> * **Q6:** "Why do you have “weather forecasting” in the title when the method is more general...?"
>
> We agree that our method is more general. We chose to emphasize "Weather Forecasting" in the title because it's the primary application in our experiments and because it motivated our work. We view the Navier-Stokes experiment as a valuable but simpler proof-of-concept, as it's a smaller-scale problem that shares similar underlying chaotic dynamics with weather.

---

> > ### Comment · Reviewer_jthi · 2025-08-03
> >
> > Thank you for your replies to my comments. My criticism has been partially addressed and I have increased my score.

---

> > > ### Author Response · Authors · 2025-08-08
> > >
> > > We are glad that we could address some of your concerns. We really appreciate your feedback, which has strengthened our revised draft.
> > >
> > > **Title**: Following your suggestion, and a similar one made by reviewer yCRd, we are planning to rename our paper to *"Elucidated Rolling Diffusion Models for Probabilistic Forecasting of Complex Dynamics"*, unless another reviewer or the AC object.

---

### Official Review · Reviewer_yCRd · 2025-06-26

**Clarity:** 3
**Significance:** 2
**Originality:** 2
**Rating:** 5
**Confidence:** 4

**Summary:**

This paper proposes an approach for modelling dynamical systems that combines the ideas from two different diffusion-based frameworks: rolling sequence diffusion models (RSDM) and elucidated diffusion models (EDM). The former models sequences, and introduces a timestep-dependent noise schedule, with increasing noise for more distant states in order to reflect the underlying uncertainty of the system. However, this approach has only been studied in the context of the DDPM formulation. Meanwhile, another popular approach emerged in the diffusion literature---EDM---which proposes carefully tuned choices for a more efficient design of the diffusion process, including noise schedules, loss weights, optimised network architectures, and faster sampling techniques. The approach proposed in this paper, ERDM, combines the strengths of the two methodologies by 1) designing a snapshot‐dependent noise schedule that assigns each future time point the appropriate noise magnitude, 2) applying a loss-weighting scheme across the sequence that emphasises the "harder" noise levels, 3) extending the backbone architecture to better capture the temporal axis, and 4) integrating efficient second-order solvers during sampling. Experiments are performed on two systems: a 2D Navier–Stokes flow simulation and the multivariate ERA5 atmospheric reanalysis dataset. The authors show that the approach generally outperforms a plain autoregressive EDM baseline (in terms of CRPS and SSR), highlighting the advantages of the rolling diffusion formulation for modelling dynamical systems.

**Questions:**

1. The authors state that it is important to tune the noise schedule parameters $\sigma_{\text{min}}$, $\sigma_{\text{max}}$, and $\rho$ during training. For example, $\rho$ is chosen such that the snapshots are under less noise than in the normal EDM formulation. How dataset-dependent are the most reasonable choices for these hyperparameters? For example, in the context of Navier-Stokes, would a different temporal resolution require re-tuning of these hyperparameters, given that the underlying dynamics change (e.g. if we use 10x a smaller timestep, the states would be much more "similar" to each other and perhaps we should add even less noise).
2. Could the authors comment on why they haven't included an RDM-based baseline too (with the DDPM formulation)?
3. Could you clarify how the sampling procedure looks like when $N$ is not an integer?
4. How much of a limitation are the increased memory and computational complexity, with the vision that such a model could be used to model, for example, higher resolution weather data (0.25° or even 0.1° ERA5)?
5. The SSR shows that ERDM goes from a region of underdispersion (up to about 10 timesteps) to overdispersion (10-20 timesteps) followed by underdispersion again. In contrast, the trend for plain EDM models is monotonically decreasing. Can the authors comment on this---what are the potential causes of overdispersion at medium timesteps, and whether the SSR could be stabilised through other hyperparameter choices?

**Ethical Concerns:**

["NO or VERY MINOR ethics concerns only"]

**Final Justification:**

As noted in my initial review, I believe the paper presents a novel combination of two previously proposed methods—Rolling Diffusion (RSDM) and EDM—adapted effectively for probabilistic spatiotemporal modelling. The paper includes comprehensive ablation studies that clearly identify which modifications lead to the most significant performance gains. Finally, the results are encouraging.
That said, I continue to believe that the authors should include a DDPM-based RSDM baseline. While the superiority of the EDM approach has been shown in the image generation literature, I think including this baseline is important for completeness and would strengthen the story of the paper. However, since I expect the proposed method to outperform the DDPM-based baseline, I have chosen not to penalise the authors heavily for this omission.
An additional comment I have after the rebuttal and reading other reviewers' comments is that the authors might stress the applicability of their approach to general probabilistic spatiotemporal tasks for chaotic systems, rather than focusing on weather forecasting right from the title. This might improve the reach and impact of their approach.

**Limitations:**

The authors mention the increased memory and computational cost of their method, but do not necessarily discuss how practical this would become with higher resolution (in space and time) data of practical significance. This would be especially relevant in the weather modelling field, where the community is heading towards 0.25° or even 0.1° spatial resolution and up to 1 hour time resolution.

**Paper Formatting Concerns:**

No paper formatting concerns.

**Quality:**

3

**Strengths And Weaknesses:**

**Strengths**
1) **Comprehensive ablations** - One of the main strengths of the paper lies in the ablations it performs. They improve the level of significance, as they help clearly identify and isolate the components that contribute most to the performance of the model.
2) **Good presentation** - The manuscript is well written, with a clear narrative flow and only minor typographical errors.
3) **Novel Integration of previous methods** - Although it does not come up with new methods, combining the RDM and the EDM formulation is original.
4) **Timely and relevant** - The topic it addresses---improving the modelling of dynamical systems with diffusion models---is of interest in the community, especially in the light of the latest advancements from communities such as weather modelling.

**Weaknesses**
1) **Variance in Loss Weighting vs. Noise Sampling** - The authors replace sampling $\sigma$ from a log-normal distribution (an importance-sampling strategy that emphasises intermediate noise levels) with an explicit loss-weighting scheme. While these approaches are equivalent in the continuous-time limit—when the loss is written as an integral over time [1]—their gradient estimates under minibatch training would differ in practice. In general, incorporating weights into the noise sampling distribution often reduces variance, so the chosen loss-weighting strategy may be suboptimal from a training perspective.
2) **First-window initialisation** - To avoid the extra hyperparameters introduced by Ruhe et al. [2], ERDM trains a separate “initial” model for the first rollout window. However, this still entails training and tuning a second network. In this work, the trained EDM model acted as a baseline anyway, but on large-scale datasets (e.g., native-resolution ERA5), maintaining two models would probably be more demanding than tuning some additional hyperparameters.
3) **Missing Baseline: Original RDM formulation** - The paper positions ERDM as an improvement over rolling diffusion, yet it does not compare directly to the original DDPM formulation of RSDM from [2]. Demonstrating gains over a DDPM-based rolling model—as well as over plain EDM—would more convincingly attribute performance improvements to the EDM design choices within the rolling framework.
4) **Experiments Setup and Benchmarks**
  - **Navier-Stokes** - Only five test trajectories are evaluated, which may not capture the full variability of initial conditions of the system. Moreover, I think the authors could include one or two additional baselines in order to be able to correctly position the performance of the method. For example, PDE-Refiner [3] showed good performance on modelling the KS and Kolmogorov systems. Another related approach is DyDiff [4]. Dyffusion does not compare to any of these, and I am not sure it's clear that Dyffusion is currently SOTA.
  - **ERA5 reanalysis** - As the authors acknowledge, comparing ERDM against NeuralGCM ENS and Graph-EFM—both evaluated on 2020—weakens the analysis due to potential distribution shifts in weather patterns during 2020/2021 (e.g., pandemic-related anomalies). Likewise, the IFS ENS comparison is flawed because IFS ENS is verified against operational analyses rather than the ERA5 reanalyses, further undermining its relevance. These mismatches leave the autoregressive EDM baseline as the only truly comparable benchmark. To strengthen their claims, the authors should: (1) include an RDM baseline following Ruhe et al. [2], and (2) consider adding a SOTA weather model—such as GenCast—to contextualise the performance gap, even if the comparison is inherently imperfect given ERDM’s coarser training data and limited computational budget. Without these additional benchmarks, it remains difficult to fully assess and contextualise the benefits of the proposed framework.

    Moreover, the results are fairly hard to interpret in the way they are currently presented. I personally think that a better way of visualising and summarising these results is through score cards.
5) **Memory Requirements** - The paper does not quantify the additional computational and memory costs of ERDM versus baseline EDM. As weather and fluid-dynamics applications move toward higher spatiotemporal resolutions, understanding ERDM’s resource footprint is crucial for assessing its practical scalability.
6) **Minor Issues**
  - **Energy Spectra** - Energy spectra would be useful in the Navier-Stokes experiment too. I am expecting a similar behaviour as in the ERA5 experiment (shown in Figure 12), but it would be worth introducing them for completeness.
- **Appendix References** - There are minor misreferences in the Appendix (e.g., the noise-prior discussion is in C.4, not C.6; the network-topology section is supposed to be in C.5., but there is no such section).
- **Sampler Efficiency Comparison** - When comparing Heun versus Euler, it would be informative to compare performance under a fixed budget of neural function evaluations (e.g., 20 Euler steps vs. 10 Heun predictor + 10 corrector steps) to account for the doubled evaluations in Heun’s corrector step.

**References**

[1] Dieleman, S. (2024). Noise schedules considered harmful. Sander.ai.

[2] Ruhe, D. et al. (2024). Rolling diffusion models. arXiv:2402.09470.

[3] Lippe, P. et al. (2023). PDE-Refiner: Achieving Accurate Long Rollouts with Neural PDE Solvers. arXiv:2308.05732.

[4] Guo, X. et al. (2025). Dynamical Diffusion: Learning Temporal Dynamics with Diffusion Models. arXiv:2503.00951.

---

> ### Author Rebuttal · Authors · 2025-07-31
>
> Thank you for your thorough and constructive review. We are grateful for your positive feedback, particularly on our *comprehensive ablations*, which you noted help to clearly identify the most impactful components of our model. We also appreciate you recognizing the paper as *well-written with a clear narrative*, and the integration of RDM and EDM as a *novel, timely, and relevant* contribution to the field.
>
> * **W1:** "Variance in Loss Weighting vs. Noise Sampling - The authors replace sampling σ from a log-normal distribution [...] with an explicit loss-weighting scheme. [...] incorporating weights into the noise sampling distribution often reduces variance, so the chosen loss-weighting strategy may be suboptimal from a training perspective."
>
> We thank the reviewer for this insightful technical point. The potential difference in gradient variance between weighting the loss versus importance-sampling the noise distribution is a valid consideration. While we acknowledge the theoretical benefits of noise sampling, we found our chosen explicit **loss-weighting** scheme to be highly effective in practice, yielding the strong empirical results reported. We will add a discussion of this trade-off, along with the suggested reference, to the section on limitations and future work.
>
>
> * **W2:** "First-window initialisation - [...] ERDM trains a separate “initial” model for the first rollout window. However, this still entails training and tuning a second network. [...] on large-scale datasets [...], maintaining two models would probably be more demanding than tuning some additional hyperparameters."
>
> This is a valid point regarding the practicality of our initialization strategy. In the context of our study, this approach was practical as a strong pretrained model (our EDM baseline) was already available. However, we acknowledge this may not be generalizable. We will explicitly state in our new *limitations section* that the reliance on a readily available, strong pretrained forecaster is a limitation of our current approach, particularly for novel problem domains.
>
> * **W3:** "Missing Baseline: Original RDM formulation - The paper [...] does not compare directly to the original DDPM formulation of RSDM from [2]. Demonstrating gains over a DDPM-based rolling model [...] would more convincingly attribute performance improvements to the EDM design choices..."
>
> This is a valid point. Our rationale for not including a direct comparison to the DDPM-based RSDM is that the *EDM framework is a direct generalization of DDPM*. As shown in the original EDM paper (c.f. Table 1), EDM recovers the DDPM formulation for specific choices of its noise schedule and preconditioning.
>
> Given the demonstrated success of the specific design choices proposed in the EDM paper, we focused our efforts on adapting these superior settings to the rolling context. While our ERDM framework inherits this generality and could be configured to replicate a DDPM-style RDM, we concentrated on what we believe to be the most promising formulation. We agree, however, that a direct empirical comparison is valuable and will note this as an important direction.
>
>
> * **W4:** "Navier-Stokes - Only five test trajectories are evaluated... Moreover, I think the authors could include one or two additional baselines... For example, PDE-Refiner [3]... Another related approach is DyDiff [4]."
>
> We thank the reviewer for these suggestions regarding the Navier-Stokes evaluation. Our use of five test trajectories follows the protocol established in the NeurIPS benchmark paper that introduced this dataset. While we agree that including additional baselines would be valuable, implementing and carefully tuning & evaluating new complex models is unfortunately hard to fulfill within the rebuttal period. We hope that the comprehensive ERA5 experiments serve as, complementary, compelling empirical evidence.
>
> * **W5:** "..., the IFS ENS comparison is flawed because IFS ENS is verified against operational analyses rather than the ERA5 reanalyses, further undermining its relevance These mismatches leave the autoregressive EDM baseline as the only truly comparable benchmark."
>
> You're right that we verify our model against *ERA5 reanalysis* while IFS ENS is verified against *operational analysis*. However, this isn't a flaw that favors our model—it's actually the opposite. This evaluation setup is the **recommended best practice** by the established WeatherBench-2 (WB2) benchmark, precisely because it's known to be more favorable to the IFS ENS baseline.
>
> An external check of the official WB2 CRPS scores for "IFS vs. Analysis" versus "IFS vs. ERA5" will confirm that the former are consistently lower (better). Therefore, our comparison is intentionally conservative. Our model's competitive performance is achieved against a baseline that is being evaluated under its most advantageous conditions. We will make sure to clarify this in the paper.
>
>
> * **W6:** "Moreover, the results are fairly hard to interpret in the way they are currently presented. I personally think that a better way of visualising and summarising these results is through score cards."
>
> That's a great suggestion, thank you. We agree that score cards are an excellent way to visualize and summarize results.
>
> Based on your feedback, we have now created a score card comparing ERDM vs. our key EDM baseline, and we will add it to the paper. Since we're not allowed to share images here, this is a high-level summary of it: The score card is predominantly "blue", indicating that ERDM outperforms EDM on the vast majority of variables and lead times, with minor exceptions for the geopotential field at 12 hours and for the q50 variable.
>
>
> * **W7:** "Memory Requirements - The paper does not quantify the additional computational and memory costs of ERDM versus baseline EDM."
>
> Excellent suggestion. We will add the following *efficiency benchmark* to our revised paper.
>
> | Model | Inference NFEs | Inference Time (s) | Inference GPU Memory (GB) | Training GPU Memory (GB) |
> |:---|:---:|:---:|:---:|:---:|
> | EDM | 600 | 237 | 21 | 19 |
> | ERDM | 120 | 209 | 49 | 53 |
>
> *Legend: Efficiency metrics for ERDM vs. EDM. All measurements were performed on a single A100 GPU using mixed precision. Inference figures correspond to a 15-day (30 time steps), 5-member ensemble forecast with second-order Heun solver. Training uses a batch size of 1.*
>
>
> * **Q1:** "How dataset-dependent are the most reasonable choices for these hyperparameters? For example, in the context of Navier-Stokes, would a different temporal resolution require re-tuning of these hyperparameters...?"
>
> In our experiments, $\rho$ was kept constant for both the Navier-Stokes and ERA5 datasets, suggesting a degree of robustness. The noise boundaries, $\sigma_{min}$ and $\sigma_{max}$, are more data-dependent, which is consistent with the standard EDM framework. We used the same $\sigma_{min}$ (0.002) for both problems, but $\sigma_{max}$ was set higher for ERA5 (500 vs. 200) to account for the larger dynamic range of its variables (e.g., humidity).
>
> Based on our results across these heterogeneous datasets, our current settings provide a robust baseline. While exploring the precise dependency of the noise schedule on data characteristics is a valuable direction for future work, and further tuning could yield marginal improvements, we found these settings to generalize well.
>
>
> * **Q2:** "Could the authors comment on why they haven't included an RDM-based baseline too (with the DDPM formulation)?"
>
> See response to **W3**. Besides, the rolling diffusion model paper [2] doesn't provide open-source code.
>
>
> * **Q3:** "Could you clarify how the sampling procedure looks like when N is not an integer?"
>
> $N$ doesn't represent a literal number of discrete solver steps. Instead, it controls the amount of denoising per step, defined as $\Delta t = 1/N$. Our sampler takes discrete steps until an internal 'global diffusion time' accumulator crosses 1, at which point the window is rolled forward.
>
> For example, when $N=1.25$, then $\Delta t = 0.8$. If the global time starts at 0.0, it takes two steps to reach 1.6, crossing the integer threshold and triggering a roll. The new global time becomes 0.6, etc. Thus, the number of actual sampling steps would always be 1 or 2 here.
>
> * **Q4:** "How much of a limitation are the increased memory and computational complexity, with the vision that such a model could be used to model, for example, higher resolution weather data (0.25° or even 0.1° ERA5)?"
>
> You're right to point out that the increased memory complexity from our 3D architecture is the primary bottleneck for scaling to higher-resolution data.
>
> We are hopeful: While standard techniques like gradient checkpointing can trade speed for memory, we believe a more powerful solution is to integrate ERDM with latent space diffusion. Performing the diffusion process on a compressed representation would drastically reduce the memory and compute costs of the 3D backbone, making the framework much more scalable without sacrificing the benefits of the rolling formulation.
>
> * **Q5:** "The SSR shows that ERDM goes from a region of underdispersion [...] to overdispersion [...] followed by underdispersion again. [...] Can the authors comment on this---what are the potential causes of overdispersion at medium timesteps, and whether the SSR could be stabilised through other hyperparameter choices?"
>
> Great observation! Note that for ERA5 (Fig. 11 in the appendix) the SSR curves are qualitatively different and this behavior doesn't happen for EDM any more. Generally, we've found that the SSR can be "improved" (for ERDM, EDM and other models) simply by stopping the training earlier. This comes at a cost of accuracy, however. Because we consider the CRPS as the key metric, we avoid doing this in this paper.

---

> > ### Comment · Reviewer_yCRd · 2025-08-02
> >
> > Thanks for your comments, I believe that including some of the additional results / comments would strengthen your paper. I have a couple more comments:
> >
> > - **W1**: "Our rationale for not including a direct comparison to the DDPM-based RSDM is that the EDM framework is a direct generalisation of DDPM. As shown in the original EDM paper (c.f. Table 1), EDM recovers the DDPM formulation for specific choices of its noise schedule and preconditioning."
> >
> >   I agree with this, and I would indeed expect your approach to outperform RDM. However, as noted earlier, the main novelty of your work lies in combining two different paradigms and adapting them to spatiotemporal modelling. For completeness, I still believe including an RDM-based benchmark would strengthen the narrative. While you mention the absence of an official codebase, as you noted yourself, it should be possible to replicate a DDPM-based setup by selecting appropriate noise schedules and preconditioning within the EDM framework.
> >
> > - **W5**: I agree that this is does not favour your model. But it might be worth explaining this in a bit more detail in the paper (in a footnote perhaps) for readers not familiar with how operational weather forecasting works.
> >
> > - **W7 and Q4**: Makes sense. Could you perhaps also include a mention of the increased memory cost in the limitations section, and how this might perhaps become a problem if you wanted to scale up to 0.25° / or even 0.1° (as in more recent efforts)? I think this is needed given that from the title you establish the focus on weather forecasting.
> >
> >   "we believe a more powerful solution is to integrate ERDM with latent space diffusion" - I agree with this point, but I do not think the community has come up with an efficient way of constructing a "good enough" latent space up until now. This is an active area of research in weather modelling. While this remains an active area of research, I’m not aware of any latent-space model that has achieved state-of-the-art performance in this domain. Thus, while the computational requirements would indeed be lowered, this route would also depend on having access to a latent space that preserves enough information in the system.
> >
> > As a broader suggestion: your title might be narrowing the potential audience. Since your approach could be applied more generally, you might consider framing it as a method for probabilistic spatiotemporal forecasting in chaotic systems, or a more concise but also more general term. This would better reflect the broader applicability beyond weather forecasting.

---

> ### Author Response · Authors · 2025-08-08
>
> Thank you again for your constructive feedback. We will certainly include these additional comments and results in our revised paper, as we believe that they significantly strengthen it.
>
> **W1 and W4**: Navier-Stokes additional baselines
>
> Upon your suggestion, we've trained two new baselines on the Navier-Stokes dataset: 1) ERDM VP, corresponding to our method, but using the VP schedule and preconditioning as in [45]; 2) PDE-Refiner [29]. PDE-Refiner excels for the first few timesteps of the rollout, but underpeforms ERDM in later stages of the rollout (being $\approx 2\times$ worse than ERDM at timestep $t=64$). We hope that these new baselines address your concerns about missing baselines. Please find a summary results table below (the first four rows correspond to a tabular version of Fig. 4 (left) in our paper).
>
> *Legend:* CRPS results for five example lead times of the test rollout (best in **bold**, second best *italized*; $\times 100$ for clarity). **New baselines:** 1) ERDM VP: Replaces the EDM-based noise schedule and preconditioning in ERDM with the corresponding variance-preserving (VP) choices (see Table 1 in [23]), which is used by the RSDM from [45]; 2) PDE-Refiner [29] is a next-step diffusion-derived forecaster. The reported scores are attained after tuning of the two key HPs of PDE-Refiner, with its final values being $K=6,\sigma_\min=0.002$. The recommended values from [29], $K=4,\sigma_\min=2e\text{-}7$, did not work as well. We use the same denoiser architecture as for EDM. PDE-Refiner performs extremely well for the first few timesteps of the rollout, but rapidly diverges after that (potentially because as a next-step forecaster, it cannot model temporal interactions).
>
> | Model | t=1 | t=8 | t=16 | t=32 | t=64 |
> | :--- | :--- | :--- | :--- | :--- | :--- |
> | DYffusion | 0.495 | 0.823 | 1.518 | 1.859 | *2.663* |
> | EDM $W=1$ | *0.381*  | 0.746 | 1.547 | 2.245 | 3.622 |
> | EDM $W=4$ | 0.406  | 0.581 | *1.165* | *1.607* | 2.811 |
> | ERDM | 0.411 | **0.394** | **0.594** | **0.932** | **1.876** |
> | ERDM VP (*new*) | 0.462  | 0.938 | 1.754 | 2.294 | 3.393 |
> | PDE-Refiner (*new*) | **0.282** | *0.503* | 1.227 | 1.919 | 3.717 |
>
> **W4:** We agree and will add this footnote, as requested.
>
> **W7 and Q4**: This is a very valid concern. While strong latent spaces are developed, simple techniques like gradient checkpointing can be used to mitigate memory concerns in the meantime (at the cost of speed). As promised, we will include a new limitations paragraph that explicitly states all limitations of our method, including the ones we discussed with you. Please find our current draft for this new paragraph below.
>
> ***Limitations.*** (new paragraph to be placed in the conclusion section of the main text)
>
> While ERDM demonstrates strong performance, its practical application is constrained by several key limitations. Its hybrid 3D denoiser architecture is computationally demanding, making each forward pass more costly in memory and speed than the 2D networks used by models like EDM. This is offset by requiring a lower number of total neural function evaluations, such that the overall sampling time is comparable to next-step EDM forecasting. The high memory complexity presents a significant barrier to scaling to higher resolutions, but could be addressed with gradient checkpointing or, potentially, latent-space diffusion. This computational cost is coupled with a notable performance weakness on ERA5 in short-range forecasts compared to state-of-the-art external baselines like IFS ENS. ERDM depends on an external forecaster for initialization, a constraint that limits its applicability in domains where a suitable model is unavailable. Lastly, ERDM relies on an explicit loss weighting to compensate for the lack of training-time noise sampling found in EDM; while this design is critical for the model's performance, it may be suboptimal in practice (Dieleman, S. (2024). Noise schedules considered harmful).
>
> **Title**: Thank you for bringing this up. Given the similar suggestion by reviewer jthi, and the generality of ERDM, we are planning to rename our paper to *"Elucidated Rolling Diffusion Models for Probabilistic Forecasting of Complex Dynamics"*, unless another reviewer or the AC object.
>
> We really appreciate all your feedback, please let us know of any remaining concerns or suggestions.

---

### Official Review · Reviewer_5Mcc · 2025-06-30

**Clarity:** 3
**Significance:** 2
**Originality:** 3
**Rating:** 4
**Confidence:** 3

**Summary:**

This paper proposes the Elucidated Rolling Diffusion Models (ERDM), a framework for probabilistic forecasting of high dimensional dynamical systems. It incorporates a loss weighting scheme, an efficient initialization strategy, and a bespoke hybrid sequence architecture to facilitate improved performance. The authors demonstrate the effectiveness of the proposed method through a series of comprehensive experiments.

**Questions:**

1. The authors should provide a clearer explanation of the rationale and applicability of diffusion models for this task.

2. In the Rolling EDM noise schedule, the authors identify the setting of the $\rho$ as particularly important. However, they only provide a simple empirical value without discussing its optimality or offering a thorough analysis.

**Ethical Concerns:**

["NO or VERY MINOR ethics concerns only"]

**Final Justification:**

The response has addressed most of my concerns.

**Limitations:**

The authors could provide a more thorough discussion of the societal limitations and potential impact of their work.

**Paper Formatting Concerns:**

There are no major formatting issues.

**Quality:**

2

**Strengths And Weaknesses:**

Paper Strengths:

1. The application of elucidated rolling diffusion models to explicitly capture the sequential nature of data and the escalating physical uncertainty inherent in chaotic systems is compelling. The authors validate the effectiveness of their approach through a comprehensive series of experiments.

2. The paper is relatively well-organized and clearly written. And it is also easy to be followed by later works.

Weaknesses:

1. The motivation of the paper appears to be unclear. It is not evident what limitations in existing methods the proposed techniques, such as the weighted loss and initialization, are intended to address. The contributions seem more like engineering refinements rather than principled innovations.

2. The authors provided some visual results in the form of charts, but there is a lack of objective quantitative tables to evaluate the performance of the proposed method compared to baseline approaches.

3. The paper lacks an analysis of its limitations. Moreover, the experiments should include measurements of inference time and computational overhead, especially considering that diffusion models often entail significant time costs during inference.

---

> ### Author Rebuttal · Authors · 2025-07-31
>
> Thank you for your review and constructive feedback. We are encouraged that you found our core approach to be "compelling". We also appreciate you recognizing that our method is validated by a comprehensive series of experiments and that the paper is well-organized, clearly written, and easy for future work to build upon.
>
> * **W1:** "The motivation of the paper appears to be unclear. It is not evident what limitations in existing methods the proposed techniques...are intended to address. The contributions seem more like engineering refinements rather than principled innovations."
>
> Thank you for this feedback, which helps us clarify the core motivation and principled nature of our work. Our approach is designed to address key limitations of existing generative methods when applied to chaotic dynamical systems, particularly in data-scarce (non Internet-scale) scenarios. Standard video diffusion models often struggle with limited data, while next-step autoregressive diffusion models may not adequately capture non-stationarity and non-Markovianity. Our ERDM framework is proposed as a *principled middle ground*, using a rolling window approach to better model the dynamics.
>
> Regarding our contributions, components like the weighted loss and initialization strategy are not mere engineering refinements but crucial, principled innovations required to make ERDM effective. For example, the weighted loss is *essential* for (1) correctly adapting the EDM training noise distribution to a sequential rolling setting and (2) is empirically vital for the model's performance, as validated by our ablation studies (Appendix, Table 1). The novelty lies in the synergistic integration of these carefully engineered components (including the noise schedule, initialization, and architecture), which are indispensable for adapting diffusion models to this challenging problem space.
>
> * **W2:** "...there is a lack of objective quantitative tables to evaluate the performance of the proposed method compared to baseline approaches."
>
> Thank you for this suggestion. While we believe our charts offer a more comprehensive view across all lead times—an approach common in the AI weather forecasting literature—we agree that a table serves as a valuable and objective reference. Therefore, for completeness, we provide the requested table below and will add it to the Appendix of our revised manuscript.
>
> **Tabular CRPS for ERA5**
>
> | Model | u10m (3d) | u10m (7d) | u10m (14d) | z500 (3d) | z500 (7d) | z500 (14d) |
> |:---|:---:|:---:|:---:|:---:|:---:|:---:|
> | EDM | 0.6923 | 1.324 | 1.769 | 76.04 | 204.9 | 327.4 |
> | **ERDM** | **0.6748** | **1.312** | **1.737** | 69.29 | 197.5 | 316.7 |
> | IFS ENS | 0.6830 | 1.318 | 1.785 | 58.41 | 183.4 | 317.3 |
> | NeuralGCM | N/A | N/A | N/A | **54.59** | **173.1** | **311.1** |
>
> *Legend: CRPS (lower is better) for 10m u-component wind (`u10m`) and 500 hPa geopotential (`z500`) at lead times of 3, 7, and 14 days.*
>
> * **W3:** *a)* "The paper lacks an analysis of its limitations." *b)* " Moreover, the experiments should include measurements of inference time and computational overhead..."
>
> *a)* Thank you for this valuable feedback. We agree that a dedicated discussion of limitations and computational overhead will significantly strengthen the paper.
>
> In our revision, we will add a new **Limitations** section to the conclusion. This section will consolidate and expand upon several points. We will reiterate the key limitation already noted in our conclusion—the higher memory footprint and cost per function evaluation of our 3D architecture compared to 2D alternatives. Furthermore, we will explicitly discuss other limitations of the current work, including:
> 1.  The scope of our ERA5 experiments, which are not performed at the highest 0.25° resolution.
> 2.  The framework's reliance on a pretrained forecaster for initialization.
>
> *b)* Excellent point. To address the need for concrete efficiency metrics, we'll include the following *efficiency benchmark* in our revised paper.
>
> | Model | Inference NFEs | Inference Time (s) | Inference GPU Memory (GB) | Training GPU Memory (GB) |
> |:---|:---:|:---:|:---:|:---:|
> | EDM | 600 | 237 | 21 | 19 |
> | ERDM | 120 | 209 | 49 | 53 |
>
> *Legend: Efficiency metrics for ERDM vs. EDM. All measurements were performed on a single A100 GPU using mixed precision. Inference figures correspond to a 15-day (30 time steps), 5-member ensemble forecast with second-order Heun solver. Training uses a batch size of 1.*
>
>
> * **Q1:** "The authors should provide a clearer explanation of the rationale and applicability of diffusion models for this task."
>
> The rationale for using diffusion models is that they often represent the state-of-the-art for probabilistic forecasting, which is essential for two primary reasons in the context of chaotic dynamical systems. First, their ability to generate *sharp, diverse ensembles* is crucial for robust *uncertainty quantification*. Second, they produce *high-fidelity samples* that are more *physically consistent* than the blurry, averaged-out predictions often generated by deterministic models trained with mean-squared error. We will add this clarification to the revised manuscript. See our *Diffusion models for spatiotemporal data* related work section for relevant references.
>
>
>
> * **Q2:** "In the Rolling EDM noise schedule, the authors identify the setting of the $\rho$ as particularly important. However, they only provide a simple empirical value without discussing its optimality or offering a thorough analysis."
>
> The insight guiding our approach is that while progressive noise is necessary, an excessively large $\rho$ destroys the temporal information that is crucial for the sequence model to learn dependencies between frames (see Fig. 2). As detailed in our ablation studies (Appendix, Table 1), the model's performance is robust across a wide range of small $\rho$ values (e.g., -5 to -30). Note that we use the same $\rho$ for Navier-Stokes and ERA5 experiments. We will add this detailed analysis to the revised manuscript.
>
>
> * **L1:** "The authors could provide a more thorough discussion of the societal limitations and potential impact of their work."
>
> We discuss the broader impact of our work in Appendix A. Is there anything specific that the reviewer is missing there?

---

> > ### Comment · Reviewer_5Mcc · 2025-08-06
> >
> > Thank you for the authors' response. The rebuttal has addressed my concerns, and I have decided to raise my score. I encourage the authors to incorporate the content from the rebuttal into the paper to improve its quality.

---

> > > ### Author Response · Authors · 2025-08-08
> > >
> > > We are glad to hear that we could address your concerns. We really appreciate your feedback, which has strengthened our revised draft, where we've incorporated the feedback from the review process. As an example, please find our current draft for the new, explicit limitations pagraph (to be placed in the conclusion section of the main text). We hope that, together with our computational complexity table above, it fully addresses your third weakness. Please let us know of any final feedback.
> > >
> > >
> > > ***Limitations.***
> > >
> > > While ERDM demonstrates strong performance, its practical application is constrained by several key limitations. Its hybrid 3D denoiser architecture is computationally demanding, making each forward pass more costly in memory and speed than the 2D networks used by models like EDM. This is offset by requiring a lower number of total neural function evaluations, such that the overall sampling time is comparable to next-step EDM forecasting. The high memory complexity presents a significant barrier to scaling to higher resolutions, but could be addressed with gradient checkpointing or, potentially, latent-space diffusion. This computational cost is coupled with a notable performance weakness on ERA5 in short-range forecasts compared to state-of-the-art external baselines like IFS ENS. ERDM depends on an external forecaster for initialization, a constraint that limits its applicability in domains where a suitable model is unavailable. Lastly, ERDM relies on an explicit loss weighting to compensate for the lack of training-time noise sampling found in EDM; while this design is critical for the model's performance, it may be suboptimal in practice (Dieleman, S. (2024). Noise schedules considered harmful).

---

### Official Review · Reviewer_yaoq · 2025-07-03

**Clarity:** 4
**Significance:** 3
**Originality:** 3
**Rating:** 4
**Confidence:** 4

**Summary:**

This paper introduces the Elucidated Rolling Diffusion Model (ERDM) that incorporates the general framework of Rolling Sequence Diffusion Models and design choices in EDM for spatial temporal probabilistic forecasts. In addition, the authors propose an uncertainty-aware weighting scheme, leverage pretrained next-step EDM models to initialize the first forecast window and design a hybrid 2D U-Net with causal temporal attention to capture spatial and temporal dependencies. ERDM outperforms baseline models on 2D Navier Stokes simulation and is competitive against state-of-the-art models on ERA5 data. With improved performance and more efficient training, ERDM promises accurate and skillful forecasts for other spatiotemporal forecasting problems.

**Questions:**

* Since the authors mentioned the inferior performance of the ERDM is due to the EDM prediction initialization, could the authors replace the current initialization with IFS ENS or  deterministic model (e.g. FourCastNet)  and repeat the experiment? to close the gap
* The noise level for the first snapshot within the window is small, does this mean it takes fewer solves to fully denoise the first snapshot? Does this have any implications of the number of NFEs compared to other models? Also, have you profiled whether the 3 D attention layers or memory bandwidth dominate? An ablation with a lighter temporal backbone would clarify where to optimize.
* How does the choice of window size and accordingly $\sigma_{\rm min}$ and $\sigma_{\rm max}$ affect the performance?
* What is the energy cost per 15 day ensemble forecast relative to IFS ENS, measured on the same hardware? This would strengthen the efficiency claim and inform practitioners.

**Ethical Concerns:**

["NO or VERY MINOR ethics concerns only"]

**Final Justification:**

I will keep my score and leaning towards acceptance. However, the methods' performance, especially in short-range skill seems to be sensitive to the initial condition but that is not evaluation and solution is address this is not provided, which can limit the applicability to other problems/areas.

**Limitations:**

Yes

**Quality:**

3

**Strengths And Weaknesses:**

Strengths:
* The proposed ERDM improves CRPS on the Navier Stokes problem compared to the best EDM and DYffuison models. Also, the ERDM is competitive with IFS ENS and NeuralCGM model with improved training efficiency.
* ERDM captures the temporal dependency and automatically account for the horizon uncertainty with the progressive noise schedule, leading to better long-term rollout.
* During sampling, the progressive noise schedule potential enables fewer solve steps than GenCast like model. After each denoised first step, the reuse of the partially denoised tail naturally moves the forecasting forward.

Weakness:
* The short-range performance on ERA5 data is inferior to IFS ENS and NeuralCGM. The authors attribute this to the less optimal initialization of the sequence. However, this needs further experiments to validate.
* It is unclear how the window size W affect the performance. Accordingly, the choice of $\sigma_{\rm min}$ and $\sigma_{\rm max}$ needs more discussion.
* the work demonstrates that diffusion models can yield calibrated, medium range probabilistic weather forecasts at modest compute cost, suggesting relevance to climate down scaling, oceanography, and other chaotic systems; however, the iterative sampling still imposes latency comparable to the EDM baseline, limiting immediate operational adoption.
* Uses existing building blocks (EDM, Heun, causal attention); conceptual novelty is incremental

---

> ### Author Rebuttal · Authors · 2025-07-31
>
> Thank you for your thoughtful and constructive review, and for recognizing our model's improved performance over strong baselines, as well as its ability to effectively capture long-range dependencies through an efficient sampling process.
>
> * **W1:** The short-range performance on ERA5 data is inferior to competitors, and the provided explanation "needs further experiments to validate."
>
> Thank you for this excellent point. We agree that our short-range performance on ERA5 data merits further discussion and that suboptimal initialization is likely only one contributing factor.
>
> We believe a more significant factor is the denoiser architecture. For both the Navier-Stokes and the ERA5 experiments, we used the same general-purpose 3D U-Net. Our method performs extremely well on the simpler Navier-Stokes problem, where designing a bespoke architecture is not as critical. However, for a complex system like ERA5, state-of-the-art performance is increasingly tied to specialized architectures (e.g., graph networks or transformers) that carefully model the spherical structure of the Earth, and its various short- and long-range dependencies. The focus of our paper was on introducing the **ERDM framework** for sequential probabilistic forecasting, rather than designing a new SOTA *architecture* for weather.
>
> We are confident that future work extending current SOTA architectures like GraphCast, Aurora, or Pangu-Weather to ingest temporal data, for use within our proposed ERDM framework, would significantly advance weather forecasting performance. We will clarify this distinction in the revised manuscript to better contextualize our results.
> * **W2:** It is "unclear how the window size $W$ affect the performance," and the choices for $\sigma_{min}$ and $\sigma_{max}$ require more discussion.
>
> Thank you for raising this important point. We have performed an ablation study on the window size $W$, $\sigma_{min}$, and $\sigma_{max}$, with the results presented in Table 1 of the Appendix. We agree that these findings should be highlighted more clearly.
>
> The ablation shows no significant sensitivity to these hyperparameters within a reasonable range. We only observed a degradation in performance for overly large window sizes, which makes the current backbone architecture less efficient for feature extraction. In our revision, we will add a sentence to the main text to explicitly summarize this and direct the reader to the detailed ablation study in the Appendix.
>
> * **W3:** Latency that limits "immediate operational adoption," despite the model's relevance to other scientific domains.
>
> Thank you for this very relevant comment on the practical implications of sampling latency. We agree with your assessment. If minimal inference latency is the single most important criterion, our model—like other iterative diffusion models such as EDM—may not be the preferred choice.
>
> In the context of operational weather forecasting, our model (like most other AI-based models) is orders of magnitude faster than incumbent physics-based systems like IFS ENS. Thus, for many applications in this domain, inference speed is not the primary limiting factor for adoption.
>
> Furthermore, this sampling speed issue could be alleviated via established techniques, such as diffusion in a compressed latent space (common in video generation), to significantly reduce the inference cost in future work.
>
> * **W4:** The conceptual novelty is "incremental" as the model uses existing building blocks.
>
> While we build upon established components, we respectfully argue that their successful integration into our ERDM framework is a non-trivial contribution tailored specifically for chaotic dynamical systems. The novelty lies in elucidating the specific adaptations required—which are not features of a standard EDM or RSDM—including a snapshot-dependent noise schedule, a bespoke temporal architecture, a specialized initialization strategy, and a tailored loss weighting.
>
> As our ablation studies demonstrate (Appendix, Table 1), these carefully engineered components are crucial for the final performance, showing that a naive combination of the building blocks would not have been sufficient.
>
> * **Q1:** "Since the authors mentioned the inferior performance of the ERDM is due to the EDM prediction initialization, could the authors replace the current initialization with IFS ENS or deterministic model (e.g. FourCastNet) and repeat the experiment? to close the gap"
>
> Thank you for this excellent suggestion for a follow-up experiment.
>
> It would certainly be possible to initialize our ERDM framework with predictions from a different model. However, we intentionally chose to use our next-step EDM baseline for initialization to ensure a carefully controlled and fair comparison. Our goal was to isolate and benchmark the specific advantages of the proposed ERDM framework against a baseline trained with the exact same setup. Using a more powerful, external model for initialization would confound the analysis, making it difficult to determine whether performance gains came from our framework or the superior initialization.
>
> As we discussed in our response to **W1**, we believe that another significant factor for the short-range performance gap is the general-purpose 3D U-Net architecture, which is not as specialized for complex weather data as SOTA architectures.
>
> * **Q2:** a) "The noise level for the first snapshot within the window is small, does this mean it takes fewer solves to fully denoise the first snapshot?" b) "Does this have any implications of the number of NFEs compared to other models?" c) "have you profiled whether the 3 D attention layers or memory bandwidth dominate? An ablation with a lighter temporal backbone would clarify where to optimize."
>
> a) Yes, the first snapshot of the first window is emitted after $N$ sampling steps (recall, $N=2$ for ERA5).
>
> b) Indeed, our ERDM framework is ***significantly more efficient in terms of the NFEs*** compared to the autoregressive EDM baseline. To generate a 30-step (15-day) forecast for our ERA5 experiments with a Euler solver, EDM requires 20 solver steps for *each* forecast, totaling $20 \times 30 = 600$ NFEs. In contrast, ERDM has a cost of only $N=2$ solver steps per forecast, resulting in just $2 \times 30 = 60$ NFEs (excluding a one-time initialization cost). These NFE counts double when using a second-order Heun solver, but the *10x* relative efficiency of ERDM is maintained.
>
> c) We experimented with a lighter 3D U-Net architecture by removing temporal attention from the shallower blocks. However, this modification resulted in negligible memory savings.
>
>
>
> * **Q3:** "How does the choice of window size and accordingly σmin and σmax affect the performance?"
>
> See response to **W2** above.
>
> * **Q4:** "What is the energy cost per 15 day ensemble forecast relative to IFS ENS, measured on the same hardware? This would strengthen the efficiency claim and inform practitioners."
>
> A direct energy comparison on the same hardware is impractical, as IFS ENS runs on specialized supercomputers, not GPUs. Nevertheless, the trend of AI models being vastly more energy-efficient is well-established; ECMWF's AI model, for example, is cited as being 1,000x more efficient than IFS ENS. Our model aligns with this trend, generating a 15-day, 5-member ensemble forecast in just 209 seconds on a single A100 GPU.

---

> > ### Comment · Reviewer_yaoq · 2025-08-06
> > **Replying to Rebuttal by Authors**
> >
> > I would like to thank the authors for the detailed discussion, clarification and alleviating most my concerns. A few question remain:
> >
> > For W1:
> > * It is claimed that the proposed model matches the ensemble skill of leading weather forecasters, but the gap in the short-range performance exists so might want to rewrite this accordingly? .
> > * "However, for a complex system like ERA5, state-of-the-art performance is increasingly tied to specialized architectures (e.g., graph networks or transformers)" - there are also recent works that show that a vanilla ViT architecture also works well for ERA5 forecasting and complex architectures might not be needed. See Nguyen, Tung, et al. "Scaling transformer neural networks for skillful and reliable medium-range weather forecasting." Advances in Neural Information Processing Systems 37 (2024): 68740-68771.
> >
> > For Q1
> > * Since it is claimed that the "inferior performance of the ERDM is due to the EDM prediction initialization",
> >  Repeating the study with a higher‑skill deterministic initializer while keeping all other settings unchanged would clarify whether the performance gap arises chiefly from the initialization or from limitations elsewhere in ERDM

---

> > > ### Author Response · Authors · 2025-08-08
> > >
> > > We are glad to hear that we could alleviate most of your concerns. We really appreciate your feedback - it has strengthened our revised draft.
> > >
> > > **W1:**
> > > -  Yes, we agree and will note this short-range underperformance on ERA5 relative to external, state-of-the-art baselines in our revised paper (e.g., this is explicitly stated in the new limitations paragraph - a full draft of it can be found in the response to reviewer yCRd)
> > > - This is a fair point. Indeed, Stormer uses a relatively straightforward architecture compared to, e.g., GraphCast. However, we note that its performance is significantly boosted by the use of a specialized "Weather Embedding" layer (see Fig. 3b of the Stormer paper), and that its ensembling technique might be a confounding factor when comparing it with raw neural networks.
> > >
> > > **Q4**: This is an excellent suggestion. While we weren't able to run this experiment for ERA5, we would like to point the reviewer to Appendix E1 and Fig. 7, where we initialize ERDM with ground truth. Since there's no higher skill initialization method, this experiment provided an upper bound on how well ERDM can *possibly* perform. Fig. 7 shows that this perfect initialization significantly improves *short-range* skill, but has no impact after the first 10 timesteps of the rollout. This indicates that ERDM's initialization is crucial for short-range skill, but insignificant for long-range skill (if a reasonably good initialization is used).

---

### Decision · Program_Chairs · 2025-09-17

**Decision:**

Accept (poster)

**Comment:**

The paper introduces *Elucidated Rolling Diffusion Models (ERDM)*, a framework for spatiotemporal probabilistic forecasting that integrates the structure of Rolling Sequence Diffusion Models with the design principles of Elucidated Diffusion Models (EDM). ERDM features a snapshot-dependent noise schedule, an uncertainty-aware loss weighting scheme, and a hybrid U-Net with causal temporal attention. It also leverages pretrained models to improve initialization for the first forecast window. The method is evaluated on two tasks: a 2D Navier–Stokes simulation and ERA5 atmospheric data, showing improvements over autoregressive EDM baselines and competitive performance compared to state-of-the-art systems like NeuralGCM and IFS ENS.

**Strengths:**
- Demonstrates superior performance over autoregressive EDM and DYffusion on Navier–Stokes, and is competitive with NeuralGCM and IFS ENS on ERA5.
- Incorporates a progressive noise schedule that better captures horizon uncertainty and enables fewer sampling steps during inference.
- Includes thorough ablation studies that isolate the contributions of each architectural and training component.
- Integrates prior methodologies (EDM and RSDM) in a novel and practical way for modelling chaotic systems.
- Paper is clearly written and well-structured, making it accessible and easy to follow.
- Shows practical training efficiency improvements and potential for broader applicability in climate and fluid dynamics modeling.

**Weaknesses:**
- Some key performance limitations remain, particularly short-range accuracy on ERA5 compared to other models, attributed to suboptimal initialization without full experimental validation.
- Lacks direct comparisons to the original DDPM-based RSDM baseline, weakening claims of methodological improvement.
- Benchmarks on ERA5 and Navier–Stokes have mismatches or limited test diversity, making it harder to fully assess generalization.
- Inference efficiency and computational overhead are insufficiently analyzed; practical deployment concerns like memory cost and inference latency are underexplored.
- Conceptual contributions may be seen as incremental engineering refinements rather than fundamentally novel ideas.
- Several empirical design choices (e.g., window size, loss weighting functions) lack principled justification or tuning analysis.

All concerns have been addressed by the authors during the rebuttal period, with all reviewers satisfied with the response. I am therefore recommending acceptance.

I would also suggest that

- The authors provide a more comprehensive literature review to include related work on diffusion models for weather forecasting like [1].
- The authors test generalization across various values of Reynolds numbers for the Navier Stokes experiments and highlight a lack of comprehensive testing across different Reynolds numbers as future work.


[1] PreDiff: Precipitation Nowcasting with Latent Diffusion Models. NeurIPS 2023.